# USE YOUR INSTINCT:
# INSTRUCTION OPTIMIZATION USING
# NEURAL BANDITS COUPLED WITH TRANSFORMERS

## ABSTRACT

Large language models (LLMs) have shown remarkable instruction-following capabilities and achieved impressive performances in various applications. However, the performances of LLMs depend heavily on the *instructions* given to them, which are typically manually tuned with substantial human efforts. Recent work has used the query-efficient Bayesian optimization (BO) algorithm to automatically optimize the instructions given to black-box LLMs. However, BO usually falls short when optimizing highly sophisticated (e.g., high-dimensional) objective functions, such as the functions mapping an instruction to the performance of an LLM. This is mainly due to the limited expressive power of the Gaussian process (GP) model which is used by BO as a surrogate to model the objective function. Meanwhile, it has been repeatedly shown that neural networks (NNs), especially pre-trained transformers, possess strong expressive power and can model highly complex functions. So, we adopt a *neural bandit* algorithm which replaces the GP in BO by an NN surrogate to optimize instructions *for black-box LLMs*. More importantly, the neural bandit algorithm allows us to naturally *couple the NN surrogate with the hidden representation learned by a pre-trained transformer* (i.e., an open-source LLM), which significantly boosts its performance. These motivate us to propose our INSTruction optimization usIng Neural bandits Coupled with Transformers (`INSTINCT`) algorithm. We perform instruction optimization for ChatGPT and use extensive experiments to show that our `INSTINCT` consistently outperforms the existing methods in different tasks, such as in various instruction induction tasks and the task of improving the zero-shot chain-of-thought instruction.

## 1 INTRODUCTION

*Large language models* (LLMs) have recently achieved remarkable performances across a variety of tasks (Zhao et al., 2023; Touvron et al., 2023). This can mainly be attributed to the strong instruction-following capability of LLMs, which allows their adaptation to various downstream applications (Liu et al., 2023; Chen et al., 2023a). However, it has been widely observed that the performances of LLMs heavily depend on the instructions/prompts given to them. These instructions are typically manually designed, which can be a human-intensive and costly process (Reynolds & McDonell, 2021; Mishra et al., 2021). Therefore, it is of paramount importance to develop efficient methods to automatically optimize the instructions/prompts to attain the best performance of LLMs. In this work, we refer to this problem as *instruction optimization* and use instructions/prompts interchangeably.

Some works have adopted gradient-based methods to optimize the instructions of LLMs (Shin et al., 2020; Li & Liang, 2021; Lester et al., 2021). However, these methods require access to the gradient of the LLMs and are hence restricted to white-box (i.e., open-source) LLMs, whereas the most powerful LLMs nowadays are typically *black-box* (e.g., ChatGPT (OpenAI, 2023a) and GPT-4 (OpenAI, 2023b)). Furthermore, even for white-box LLMs, gradient computation becomes resource-intensive and hence less practical as the models become larger, which is another limitation of gradient-based methods. Therefore, recent works have proposed instruction optimization methods not requiring the model gradient, which are able to optimize the instructions for black-box LLMs (Zhou et al., 2023; Prasad et al., 2022; Pryzant et al., 2023). However, these methods are based on heuristic local search and are hence not able to leverage the observation history (i.e., the previously queried instructions

and their scores) when selecting new instructions to query. As a consequence, these methods are not able to balance *exploration* of the entire space of instructions (to query instructions whose scores are uncertain) vs. *exploitation* of the current observation history (to query instructions predicted to have high scores based on the observation history). This makes them *query-inefficient* and hence impractical when the API calls to black-box LLMs incur costs such as monetary and time expenses.

In this regard, the recent work of Chen et al. (2023b) has proposed the InstructZero algorithm which optimizes the instruction using the *query-efficient* Bayesian optimization (BO) algorithm (Garnett, 2023). To apply BO, InstructZero uses *a separate white-box LLM* to convert instruction optimization for black-box LLMs to a continuous optimization problem, i.e., optimizing the *soft prompt* which is a continuous vector (more details in Sec. 2.1). Then, InstructZero uses a Gaussian process (GP) (Rasmussen & Williams, 2006) as a sur-

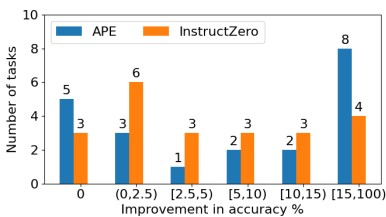

Figure 1: Improvements of our `INSTINCT` over baselines (in 30 tasks).

rogate to model the objective function (i.e., the function mapping a soft prompt to a score), and sequentially selects the soft prompts to query by maximizing an acquisition function which balances exploration and exploitation in a theoretically grounded manner. However, it has been shown that BO often falls short when optimizing highly sophisticated or high-dimensional objective functions (Dai et al., 2022), such as the function mapping a soft prompt to the performance (i.e., score) of an LLM. This important shortcoming of BO is mainly attributed to the limited expressive power of the GP surrogate. On the other hand, it has been repeatedly shown that *neural networks* (NNs), especially pre-trained *transformer* models (Vaswani et al., 2017), possess strong expressive power and can model highly complex functions with high-dimensional inputs.

Therefore, in this work, we perform instruction optimization for black-box LLMs by adopting a recently developed *neural bandit* algorithm: *Neural Upper Confidence Bound* (NeuralUCB) (Zhou et al., 2020) (Sec. 2.2). NeuralUCB replaces the GP surrogate in BO with an NN surrogate while preserving the ability of BO to trade-off exploration vs. exploitation in a principled way. More importantly, NeuralUCB allows us to naturally *couple the NN surrogate with the hidden representation learned by a pre-trained transformer* (i.e., a white-box LLM), which further improves the capability of the NN surrogate for score prediction and hence boosts the performance of our algorithm. As a result, we propose our INSTruction optimization usIng Neural bandits Coupled with Transformers (`INSTINCT`) algorithm (Sec. 3). In our empirical evaluations, we optimize the instructions for the black-box LLM ChatGPT (OpenAI, 2023a) and adopt Vicuna (Chiang et al., 2023) as the white-box LLM. We use extensive experiments to show that our `INSTINCT` consistently outperforms the existing methods in different tasks (Sec. 4), such as in various instruction induction tasks (Sec. 4.1) and the task of improving the zero-shot chain-of-thought (CoT) instruction (Sec. 4.2). We also use ablation studies to unveil interesting insights about our `INSTINCT` algorithm (Sec. 5).

## 2 BACKGROUND AND PROBLEM SETTINGS

### 2.1 BAYESIAN OPTIMIZATION FOR INSTRUCTION OPTIMIZATION

**Instruction Optimization.** A black-box LLM $f$ takes as input an *instruction* $\rho$ prepended to a test input $x$, and outputs a sentence $\hat{y} = f(\rho, x)$. The LLM $f$ is *black-box* in that we can only query it via its API and cannot access its parameters. We consider a language task with a validation dataset $D_V = \{(x_i, y_i)\}_{i=1}^n$ of $n$ pairs of input sentence $x_i$ and its corresponding ground truth output sentence $y_i$. For an instruction $\rho$ and an input $x_i$, a score function $s(\cdot, \cdot)$ compares the LLM output sentence $\hat{y}_i = f(\rho, x_i)$ with the ground truth output sentence $y_i$ to return a score $s(\hat{y}_i, y_i)$. As a result, instruction optimization can be formulated as the problem of finding the optimal instruction $\rho^*$ that achieves the highest score averaged over the validation set $D_V$. Note that the performance of the instruction $\rho$ we find using the validation set $D_V$ is evaluated using a separate test set $D_T$.

Directly optimizing the instruction $\rho$ for a black-box LLM $f$ is challenging because of the combinatorial nature of the tokens forming the instruction $\rho$. To this end, InstructZero (Chen et al., 2023b) has used a separate white-box (i.e., open-source) LLM $w$ to convert this combinatorial optimization problem (i.e., optimizing $\rho$) into continuous optimization, i.e., optimizing a *soft prompt* $z$. Specifically, a soft prompt $z \in Z \subset \mathbb{R}^d$ is a $d$-dimensional continuous vector and corresponds to the token embeddings of a number $N_z$ of soft tokens (Lester et al., 2021). A soft prompt $z$ is prepended to

the token embeddings of a fixed set $E$ of input-output exemplars $E = \{(x_\tau, y_\tau)\}_{\tau=1}^{\kappa}$ for the task. These concatenated embeddings are used as the input to the white-box LLM $w$, which subsequently generates an instruction $\rho(z) = w(z, E)$. The generated instruction $\rho(z)$ is then prepended to a test input $x_i$ (from the validation dataset $D_V = \{(x_i, y_i)\}_{i=1}^{n}$) and used as input to the black-box LLM $f$ to generate an output sentence $\hat{y}_i = f(\rho(z), x_i)$, which is then evaluated to produce a score $s(\hat{y}_i, y_i)$. In doing so, with a fixed set $E$ of exemplars, the discrete optimization problem of optimizing $\rho$ is converted to the optimization of a continuous soft prompt $z$:

$$z^* = \arg\max_{z \in Z} h(\rho(z)), \quad h(\rho(z)) \triangleq \mathbb{E}_{(x,y) \in D_V} s(f(\rho(z), x), y) = (1/n)\sum_{i=1}^{n} s(\hat{y}_i, y_i). \quad (1)$$

Based on this formulation, InstructZero (Chen et al., 2023b) has adopted *Bayesian optimization* (BO) (Garnett, 2023) to maximize the objective function $h(\rho(z))$ (equation 1). To achieve this, a *Gaussian process* (GP) (Rasmussen & Williams, 2006) is used as a surrogate to model the function $h(\rho(z))$. In every iteration $t$ of BO, the current observation history is used to update the GP model which is then used to calculate an *acquisition function* $\alpha_t(z)$. Then, a soft prompt $z_t$ is selected by maximizing $\alpha_t(z)$: $z_t = \arg\max_{z \in Z} \alpha_t(z)$. Next, the selected $z_t$ is used as input to the white-box LLM $w$ to produce an instruction $\rho_t$, which is then evaluated using the black-box LLM $f$ to produce a score $h_t$ (details in Sec. 3.3). Lastly, the newly collected input-output pair $(z_t, h_t)$ is added to the observation history to update the GP model, which is then used to select the soft prompt $z_{t+1}$ in the next iteration.

The soft prompt $z$ is normally high-dimensional (e.g., $d = 5120 \times N_z$ when $w$ is Vicuna 13B), which makes it challenging for BO to optimize. So, InstructZero (Chen et al., 2023b) has adopted the technique of *random projection* to reduce the input dimension. That is, given a matrix $A \in \mathbb{R}^{d \times d'}$ ($d' \ll d$) with randomly sampled elements and a $d'$-dimensional continuous vector $\hat{z}$, the vector $z = A\hat{z}$ is used as the soft prompt. After substituting the $z$ by $A\hat{z}$, the input variable to be optimized in equation 1 is changed to $\hat{z}$ and hence the input dimension of the optimization problem is reduced to $d'$. The reduced input dimension $d'$, i.e., the *intrinsic dimension*, is chosen as $d' = 10$ in InstructZero.

## 2.2 NEURAL BANDITS

Neural bandit algorithms, such as NeuralUCB (Zhou et al., 2020) we have adopted in this work, replace the GP surrogate in BO (Sec. 2.1) by a neural network (NN) while preserving the principled ability of BO to trade-off exploration vs. exploitation. The strong expressive power of NNs equips neural bandit algorithms with the ability to optimize highly complicated objective functions, which is theoretically justified (Dai et al., 2022). In practice, neural bandit algorithms have also been shown to outperform BO especially in problems with sophisticated objective functions (Lisicki et al., 2021). However, naively applying NeuralUCB to our problem is challenged by the huge computational costs. This is because every evaluation of the NeuralUCB acquisition function requires performing an inference using the white-box LLM, which can be extremely costly since the acquisition function needs to be evaluated many times in every iteration. So, we use a technique based on pre-computation (Sec. 3.2) to sidestep this expensive computation and hence make our `INSTINCT` algorithm scalable. Moreover, we also couple the NN surrogate in NeuralUCB with the powerful hidden representation learned by a pre-trained transformer to further improve the performance of our `INSTINCT` (Sec. 3.1).

## 3 INSTINCT ALGORITHM FOR INSTRUCTION OPTIMIZATION

**Overview.** In every iteration $t$ of our `INSTINCT` algorithm (Fig. 2), we firstly use the current observation history (i.e., pairs of soft prompts and observed scores) to train an NN for score prediction (**step** ①, Sec. 3.1), and use the trained NN to calculate the NeuralUCB acquisition function (equation 2), which is then maximized to select the next soft prompt $z_t$ to query (**step** ②, Sec. 3.2). Next, we feed the selected $z_t$ (and a small set $E$ of exemplars for the task) as the input to the white-box LLM, which then generates an instruction $\rho_t$ (**step** ③). Then, $\rho_t$ is evaluated using the validation set $D_V$ (**steps** ④ and ⑤), which produces a score $h_t$. In the subsequent sections, We discuss every step of our `INSTINCT` in the following sections, with some technical details deferred to App. E.

## 3.1 TRAINING NEURAL NETWORK FOR SCORE PREDICTION (STEP ①)

In **step** ①, we use the current observation history to train an NN, i.e., a multi-layer perceptron (MLP), for score prediction. Here we use $g$ to denote the mapping from a soft prompt $z$ to its corresponding *hidden representation of the last token in the final layer* of the pre-trained transformer (i.e., the

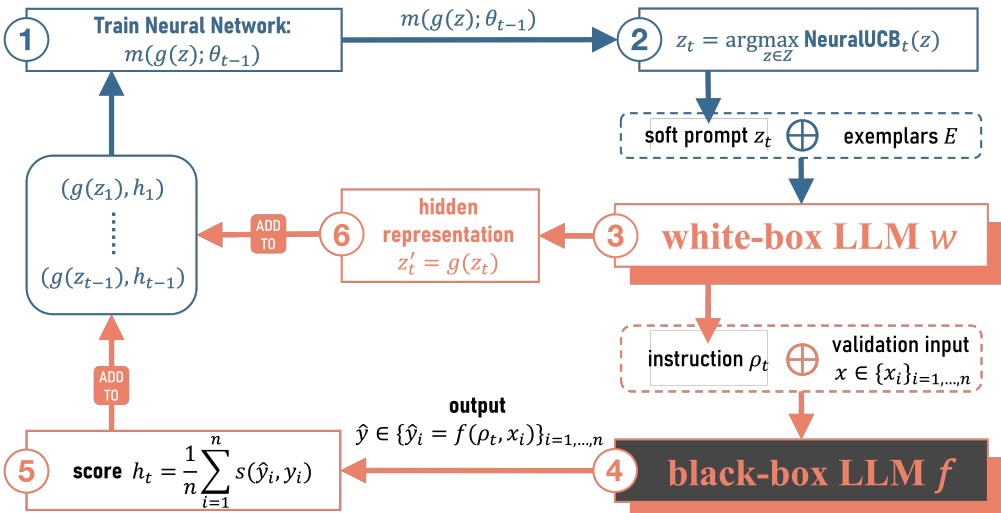

Figure 2: Illustration of our INSTINCT algorithm. Every step is described in detail in Sec. 3.

white-box LLM $w$): $z' = g(z)$.[1] Hereafter, we refer to $z' = g(z)$ as **the hidden representation** of $z$ for simplicity. Importantly, thanks to the strong expressive power of the pre-trained transformer, we can **stack an NN (i.e., MLP) on top of the hidden representation** $z' = g(z)$ to achieve accurate score predictions (more details below). Also note that connecting the hidden representation (of the last token in the final layer) of a pre-trained transformer with an MLP to perform prediction tasks has been commonly adopted and proven effective in various applications (Radford et al., 2018; 2019).

We use $m(g(z); \theta)$ to denote an NN (i.e., MLP) with parameters $\theta$ and input hidden representation $z' = g(z)$. Note that although our NN $m(g(z); \theta)$ is used to predict a score for every soft prompt $z$, we have used $z' = g(z)$ to represent its input because we freeze the parameters of the pre-trained transformer model, i.e., the hidden representation $z' = g(z)$ for every $z$ is fixed. In iteration $t$, given the first $t - 1$ observations $\{(g(z_\tau), h_\tau)\}_{\tau=1}^{t-1}$, we train our NN $m(g(z); \theta)$ using Adam (Kingma & Ba, 2014) to minimize the mean squared error (MSE) loss with an L2 regularization parameter $\lambda$. This yields the updated NN parameters $\theta_{t-1}$. Importantly, the resulting NN $m(g(z); \theta_{t-1})$ can both leverage the powerful hidden representation $z' = g(z)$ learned by the pre-trained white-box LLM and adapt to the task of score prediction thanks to NN training. So, the trained NN $m(g(z); \theta_{t-1})$ is able to *accurately predict the scores of different soft prompts*, which is crucial for the compelling performance of our INSTINCT algorithm. After the NN training, we use it to select the next soft prompt to query via the NeuralUCB acquisition function, which we discuss in the next section.

### 3.2 Selecting the Next Soft Prompt $z_t$ (Step ②)

In **step ②**, we choose the next soft prompt $z_t$ to query by maximizing the NeuralUCB acquisition function (Sec. 2.2). Specifically, we use the trained NN $m(g(z); \theta_{t-1})$ (Sec. 3.1) to calculate the acquisition function value $\text{NeuralUCB}_t(z)$ for every soft prompt $z \in Z$ in the domain, which is then maximized across all $z \in Z$ to choose the next soft prompt $z_t$ to query:

$$z_t = \arg\max_{z \in Z} \text{NeuralUCB}_t(z), \quad \text{NeuralUCB}_t(z) \triangleq m(g(z); \theta_{t-1}) + \nu_t \sigma_{t-1}(g(z); \theta_{t-1}), \quad (2)$$

in which $m(g(z); \theta_{t-1})$ denotes the predicted score for soft prompt $z$, $\sigma_{t-1}(g(z); \theta_{t-1})$ is a *principled measure of our uncertainty* about the function value $h(z)$ at $z$ which is calculated using the gradient of the NN (Jacot et al., 2018) (see its detailed expression in App. E), and $\nu_t$ is a weighting parameter. As a result, the acquisition function $\text{NeuralUCB}_t(z)$ (equation 2) is able to select a soft prompt $z_t$ by simultaneously encouraging both **(i) exploitation** of the observation history $\{(g(z_\tau), h_\tau)\}_{\tau=1}^{t-1}$ to encourage the selection of soft prompts predicted to have high scores $m(g(z); \theta_{t-1})$ and **(ii) exploration** of entire domain $Z$ of soft prompts by preferring the selection of soft prompts with larger uncertainty $\sigma_{t-1}(g(z); \theta_{t-1})$. Intuitively, we are able to both **(i)** leverage the accurate score prediction enabled by the strong expressivity of the NN coupled with the powerful hidden representation learned

---

[1]Note that in addition to the soft prompt $z$, the hidden representation $z'$ also depends on other factors such as the set $E$ of exemplars. We have omitted the dependency on these factors here as they are kept fixed for a task.

by the pre-trained transformer $w$ and **(ii)** perform principled exploration of the entire domain of soft prompts thanks to the principled uncertainty measure $\sigma_{t-1}(g(z); \theta_{t-1})$, which combine to lead to the strong practical performance of our INSTINCT algorithm (Sec. 4). We defer more detailed explanations of NeuralUCB$_t(z)$ (equation 2) to App. E.

**Pre-Computation to Save Costs.** Interestingly, since our INSTINCT algorithm does not require updating the pre-trained hidden representation $z' = g(z)$, we can adopt a natural technique to significantly reduce its computational cost. That is, before running our INSTINCT, we generate a discrete domain $\widetilde{Z}$ of soft prompts (details in the next paragraph) using a scrambled Sobol sequence following the common practice in BO (Eriksson et al., 2019), which ensures that the discrete domain $\widetilde{Z}$ has a good coverage of the original continuous domain $Z$. Then, we **pre-compute** the hidden representation $z' = g(z)$ for every soft prompt $z$ in this discrete domain $\widetilde{Z}$. Given all pre-computed hidden representations (i.e., $z' = g(z)$ for all $z \in \widetilde{Z}$), when selecting the next soft prompt $z_t$ during our algorithm (equation 2), we can instead maximize over the fixed discrete domain $\widetilde{Z}$: $z_t = \arg\max_{z \in \widetilde{Z}} \text{NeuralUCB}_t(z)$. This considerably reduces the computational cost because every hidden representation $g(z)$ (used in the calculation of NeuralUCB$_t(z)$) has been pre-computed.

**Generating the Discrete Domain $\widetilde{Z}$.** We have also adopted the technique of *random projection* (Sec. 2.1) when generating the discrete domain $\widetilde{Z}$. Specifically, instead of directly generating a scrambled Sobol sequence of $d$-dimensional vectors ($d$ is the dimension of the soft prompt $z$), we generate a sequence of $d'$-dimensional vectors ($d' \ll d$), which constitute a discrete domain in the $d'$-dimensional space, denoted as $\widetilde{Z}'$ (Sec. 2.1). Next, we use a matrix $A \in \mathbb{R}^{d \times d'}$ (with randomly sampled elements) to project every point $\hat{z} \in \widetilde{Z}'$ in the $d'$-dimensional discrete domain $\widetilde{Z}'$ to the original $d$-dimensional space: $z = A\hat{z}$ for all $\hat{z} \in \widetilde{Z}'$. The resulting projected $d$-dimensional vectors constitute our discrete domain $\widetilde{Z}$. We have adopted this technique of random projection to generate $\widetilde{Z}$ because it provides us a simple way to adjust the overall magnitudes of the soft prompts in the discrete domain $\widetilde{Z}$ by tuning the intrinsic dimension $d'$. Intuitively, a larger intrinsic dimension $d'$ in general causes the soft prompts $z \in \widetilde{Z}$ to have larger magnitudes/norms (see detailed explanation in App. E.2), and different tasks may be suitable for soft prompts with different overall magnitudes. Therefore, this flexibility to choose $d'$ allows us to *automatically adapt to the task at hand* by using the validation set to tune $d'$ and hence further boosts the performance of our INSTINCT algorithm.

### 3.3 EVALUATING THE SELECTED SOFT PROMPT $z_t$ (STEPS ③-⑤)

After the soft prompt $z_t$ is selected (Sec. 3.2), we proceed to evaluate its performance. Specifically, the selected soft prompt $z_t$ is prepended to the embeddings of a set $E$ of exemplars (as well as other texts such as "The instruction was to"), and then the concatenated embeddings are used as the input to the white-box LLM $w$ to generate an instruction $\rho_t = w(z_t, E)$ (**Step ③**). Next, for every input $x_i$ in the validation set $D_V = \{(x_i, y_i)\}_{i=1}^n$, we prepend the generated instruction $\rho_t$ to $x_i$ and then use them as the input to the black-box LLM $f$ to generate its output sentence $\hat{y}_i = f(\rho_t, x_i)$ (**Step ④**), which is used to calculate a score $s(\hat{y}_i, y_i)$. The score $h_t$ for $\rho_t$ is therefore calculated by averaging over the validation set: $h_t = (1/n) \sum_{i=1}^n s(\hat{y}_i, y_i)$ (**Step ⑤**). Lastly, we extract the hidden representation of $z_t$: $z_t' = g(z_t)$ (**Step ⑥**), and add the newly collected input-output pair $(g(z_t), h_t)$ to the observation history, which is subsequently used to train the NN $m(g(z); \theta_t)$ (**Step ①**) and then select the soft prompt $z_{t+1}$ in the next iteration (**Step ②**).

### 3.4 STRENGTHS OF OUR INSTINCT

The strengths of our INSTINCT lie in not only its **enhanced exploitation** (i.e., accurate score prediction) facilitated by our NN surrogate and its coupling with the hidden representation from the pre-trained transformer (discussed in Sec. 3.1), but also its **better exploration** enabled by our principled uncertainty estimation. In particular, the exploration of BO/neural bandits relies on a good *similarity measure* between different pairs of soft prompts. That is, if a pair of soft prompts leads to similar scores (i.e., function values), a reliable similarity measure should assign a large similarity value to this pair of soft prompts. However, an important challenge faced by the framework adopted by both InstructZero and our INSTINCT (Fig. 2) is that *different soft prompts can lead to the same instruction and hence the same score* (we verify this in Sec. 5), and these pairs of soft prompts should be given large similarity values. Unfortunately, InstructZero cannot effectively handle this

Table 1: Average test accuracy (standard error) achieved by the best instruction discovered by different algorithms for different tasks (3 independent trials with different random seeds). For better distinguishability, only the tasks for which any method has an average test accuracy less than $0.8$ (i.e., more challenging tasks) are included. The results including all tasks are given in Table 9 (App. C.2).

| Task | APE | InstructZero | INSTINCT (ours) |
|------|-----|--------------|-----------------|
| antonyms | 0.6367(0.1416) | 0.8267(0.0072) | **0.8467(0.0027)** |
| auto_categorization | 0.2500(0.0094) | **0.2567(0.0119)** | 0.2500(0.0330) |
| auto_debugging | 0.2917(0.0340) | **0.3750(0.0000)** | 0.2917(0.0340) |
| cause_and_effect | 0.5733(0.0891) | **0.8133(0.0109)** | 0.5867(0.0871) |
| common_concept | 0.0691(0.0207) | 0.0864(0.0398) | **0.2129(0.0019)** |
| diff | 0.6733(0.2667) | 0.6933(0.2224) | **1.0000(0.0000)** |
| informal_to_formal | **0.5736(0.0026)** | 0.5310(0.0024) | 0.5534(0.0000) |
| letters_list | **1.0000(0.0000)** | 0.5900(0.1674) | **1.0000(0.0000)** |
| negation | 0.7533(0.0109) | 0.7767(0.0136) | **0.8167(0.0027)** |
| object_counting | **0.3633(0.0191)** | 0.3600(0.0929) | 0.3400(0.0698) |
| odd_one_out | 0.6333(0.0144) | 0.6133(0.0871) | **0.7000(0.0163)** |
| orthography_starts_with | 0.4567(0.1477) | 0.5067(0.0871) | **0.6667(0.0272)** |
| rhymes | 0.1567(0.0640) | **1.0000(0.0000)** | **1.0000(0.0000)** |
| second_word_letter | **0.7467(0.2028)** | 0.4333(0.1872) | 0.1000(0.0411) |
| sentence_similarity | 0.0000(0.0000) | 0.0000(0.0000) | **0.1400(0.0047)** |
| sum | 0.6733(0.2667) | **1.0000(0.0000)** | **1.0000(0.0000)** |
| synonyms | **0.3600(0.0759)** | 0.2767(0.0925) | 0.3067(0.0491) |
| taxonomy_animal | 0.3467(0.2341) | 0.7167(0.0838) | **0.8567(0.0599)** |
| word_sorting | 0.3300(0.0374) | 0.3100(0.1143) | **0.5133(0.0027)** |
| word_unscrambling | 0.4400(0.1389) | 0.5500(0.0170) | **0.6333(0.0072)** |
| # best-performing tasks | 5 | 5 | 13 |
| # second-best-performing tasks | 5 | 10 | 5 |
| average rank | 2.25 | 2.0 | 1.45 |

issue because it has made use of standard similarity measures from BO (i.e., the Matérn kernel) to measure similarity in the *original* space of soft prompts.[2] In contrast, our INSTINCT can better resolve this issue thanks to the use of the hidden representation in our principled uncertainty measure $\sigma_{t-1}(g(z); \theta_{t-1})$ (equation 2): if two soft prompts lead to the same instruction (and hence the same score), the distance between their hidden representations is also small. We have empirically verified this in our ablation study (Sec. 5). Therefore, the superiority of our INSTINCT in terms of both **exploitation** and **exploration** helps it achieve consistently better performances than existing methods.

## 4 EXPERIMENTS

We perform instruction optimization for ChatGPT and use Vicuna-13B as the white-box LLM $w$. We conduct instruction induction tasks using 30 datasets from Chen et al. (2023b) (Sec. 4.1), and the task of improving the zero-shot chain-of-thought instruction using 3 arithmetic reasoning datasets: GSM8K (Cobbe et al., 2021), AQUARAT (Ling et al., 2017) and SVAMP (Patel et al., 2021) (Sec. 4.2). We compare our INSTINCT with two representative baselines: **APE** (Zhou et al., 2023) and **InstructZero** (Chen et al., 2023b). Following InstructZero, we initialize our algorithm by randomly selecting 40 soft prompts, and then run our INSTINCT to query another 125 soft prompts. For both our INSTINCT and InstructZero, for all tasks unless specifically specified, we use the validation set $D_V$ to perform a grid search over the intrinsic dimension $d'$ in $\{10, 50, 100\}$ and the number $N_z$ of soft tokens in $\{3, 5, 10\}$ (Sec. 2.1). This ensures that our INSTINCT use the same the total number of queries to the black-box LLM as InstructZero for a fair comparison. For every algorithm, after finding the best instruction using the validation set $D_V$, we evaluate the discovered instruction using the separate test set $D_T$ and report the test accuracy as the score. More details on the experiments are deferred to App. C.1.

### 4.1 INSTRUCTION INDUCTION

Here we aim to find a task-specific instruction that best describes the relationship between inputs and outputs of a given task. We report in Table 1 the test accuracy achieved by the best instruction discovered by different methods for various instruction induction tasks. Our INSTINCT achieves the highest accuracy in 13 out of the 20 tasks, with an average rank of $1.45$ which is significantly

---

[2] The instruction-coupled kernel used by InstructZero (Chen et al., 2023b) can only resolve this issue when measuring the similarity between a pair of *already-queried* soft prompts. See detailed explanations in App. D.1.

| Method | Dataset | Best Zero-Shot CoT Instruction | Score |
|---|---|---|---|
| Kojima et al. (2022) | GSM8K | Let's think step by step. | 0.71797 |
| InstructZero | GSM8K | Let's use the instruction to solve the problem. | 0.74299 |
| INSTINCT (ours) | GSM8K | **Let's think about it.** | **0.74526** |
| Kojima et al. (2022) | AQUA-RAT | Let's think step by step. | 0.52362 |
| InstructZero | AQUA-RAT | Let's break down the problem. | 0.54331 |
| INSTINCT (ours) | AQUA-RAT | **I have a new solution.** | **0.54724** |
| Kojima et al. (2022) | SVAMP | Let's think step by step. | 0.7625 |
| InstructZero | SVAMP | Let's use the equation. | 0.795 |
| INSTINCT (ours) | SVAMP | **Let's use our brains.** | **0.81** |

Table 3: The best zero-shot CoT instructions found by different algorithms and their scores.

better than APE and InstructZero. We have shown these 20 tasks here because they allow for better distinguishability among different algorithms, and the advantage of our INSTINCT is consistent in the table including all 30 tasks (Table 9 in Appendix). We have also performed a text summarization task using the SAMSum dataset (Gliwa et al., 2019) (Table 2), in which our INSTINCT again performs the best. The results here demonstrate the superior capability of our INSTINCT for instruction optimization across a variety of tasks. In the Appendix, we have also illustrated how our INSTINCT algorithm is able to generate higher-quality instructions across different iteration in Fig. 8, and presented the best final instruction our INSTINCT discovered for every task in Table 8.

## 4.2 IMPROVING ZERO-SHOT CHAIN-OF-THOUGHT PROMPT

Chain-of-thought (CoT) reasoning has been found to be an effective technique to boost the performance of LLMs in complex tasks that require multiple steps of reasoning (Wei et al., 2022). The work of Kojima et al. (2022) has discovered that the performance of LLMs in complicated reasoning tasks can be significantly improved by simply prepending the *zero-shot CoT instruction* "Let's think step

Table 2: Instruction optimization on SAMSum dataset (summarization task).

| Method | ROUGE-1 | ROUGE-2 | ROUGE-L |
|---|---|---|---|
| APE | 0.32549 | 0.10308 | 0.30245 |
| InstructZero | 0.32595 | 0.10528 | 0.30061 |
| INSTINCT | **0.35580** | **0.13350** | **0.33600** |

by step." to the questions, which outperforms other manually designed instructions. Here we show that our INSTINCT algorithm can further improve over this zero-shot CoT instruction across multiple tasks in Table 3. We defer our detailed experimental design to App. C.3.

## 5 ABLATION STUDY

**Effectiveness of the Hidden Representation.** Here we empirically verify that the use of the hidden representation from the pre-trained transformer when building the NN surrogate (Sec. 3.1) indeed helps improve the performance of our INSTINCT algorithm. To this end, we compare the evolution of the performances of our INSTINCT algorithm with and without using the hidden representation across different iterations. The results in Fig. 3 suggest that in the early iterations with a small number of observations, the use of the hidden representation allows our NN surrogate to quickly learn to accurately predict the scores and hence helps our INSTINCT algorithm quickly achieve high accuracies. After more iterations, our INSTINCT algorithm without using the hidden representation can also achieve competitive performances after enough observations (i.e., training data) have been collected such that the NN surrogate can be trained to accurately predict the scores.

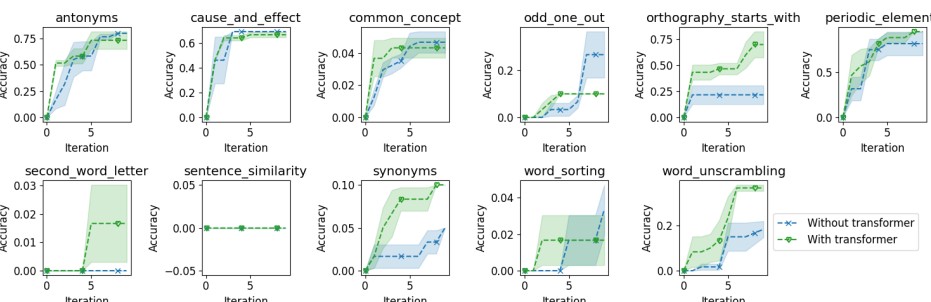

Figure 3: The performance of best prompts in early iterations. The tasks plotted are those for which the performance gap is larger than 0.01 with and without using the hidden representation.

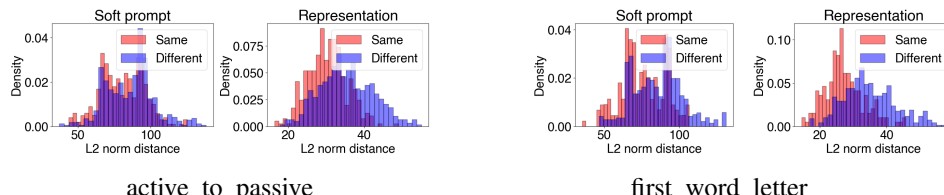

active_to_passive      first_word_letter

Figure 4: Pairwise L2 distances between soft prompts (left) and hidden representations (right) for soft prompts mapping to the same (red) and different (blue) instructions. See details in Sec. 5.

**Hidden Representation Give Better Similarity Measure.** As discussed in Sec. 3.4, an important challenge faced by both InstructZero (Chen et al., 2023b) and our `INSTINCT` is that *multiple soft prompts can lead to the same instruction and hence the same score*, and this issue cannot be effectively handled by InstructZero (Chen et al., 2023b) (more explanations in App. D.1).[2] In contrast, our `INSTINCT` can better resolve this issue because our principled uncertainty measure is calculated *based on the hidden representations*. For two soft prompts mapping to the same instruction, the distance between their hidden representations is small. Here we empirically verify this and show the results in Fig. 4 (more results in Fig. 10, App. D.1). We firstly construct 2 groups of soft prompts: the soft prompts in the first group map to the same instruction (referred to as the "Same" group), and the soft prompts in the second group map to different instructions (the "Different" group). For both the "Same" group (red color in Fig. 4) and "Different" group (blue color in Fig. 4), for *every pair of soft prompts within a group*, we compute the pairwise L2 distance between both the original soft prompts (left figure for each task in Fig. 4) and their hidden representations (right figure). The right figure for each task (Fig. 4) show that **the pairwise distances between the hidden representations within the "Same" group (red) are markedly smaller than those for the "Different" group (blue)**; meanwhile, this notable difference between the 2 groups is not observed in the left figure for each task (i.e., the pairwise distances between original soft prompts). This indicates that the hidden representation make it significantly easier to resolve the above-mentioned issue, and hence our `INSTINCT` can perform better exploration. We also use another ablation study (Table 7, App. D.4) to verify that our principled exploration is necessary for the competitive performance of our `INSTINCT`.

**Improving `INSTINCT` via One-shot In-Context Learning.** Here we show that the performance of our `INSTINCT` can be further improved via in-context learning (ICL) (Brown et al., 2020), which is a widely used method to boost the performance of LLMs. We propose two methods to incorporate ICL into our `INSTINCT`: **(i)** *test-time-only one-shot `INSTINCT`* which only appends an exemplar after *the best instruction discovered* by our `INSTINCT` algorithm (and pass the concatenated instruction-exemplar to the black-box LLM for evaluation) at test time, and **(ii)** *one-shot `INSTINCT`* which appends an exemplar after *every queried instruction $\rho_t$* during our `INSTINCT` algorithm. The results (Table 4) show that adding an exemplar to the best-discovered instruction (test-time-only one-shot `INSTINCT`) improves the performance of our `INSTINCT`. Additionally adding the one-shot exemplar to every queried instruction during our `INSTINCT` (one-shot `INSTINCT`) further enhances the performance, which is likely because this improves the alignment between our optimization objective and test performance. These results demonstrate the compatibility of our `INSTINCT` with ICL and suggest wider potential applications of our `INSTINCT` through its combination with ICL.

**Improving `INSTINCT` with ChatGPT Rephrasing.** Here we propose a technique to further improve the performance of our `INSTINCT` based on the resampling technique from APE (Zhou et al., 2023). Specifically, in every iteration after the instruction $\rho_t$ is generated by the white-box LLM (Fig. 2), instead of directly passing $\rho_t$ to the black-box LLM for evaluation, we firstly pass $\rho_t$ to ChatGPT and instruct it to rephrase and improve this instruction $\rho_t$ to obtain a new instruction $\rho_t'$. Then, the new instruction $\rho_t'$ is evaluated by the black-box LLM to produce the score $h_t$. We have applied this improved variant of our `INSTINCT` algorithm to those instruction induction tasks (Sec. 4.1) with large room for improvement, i.e., the tasks with average test accuracy below 0.8 (Table 1). We plot the histogram of the improvements (i.e., improved average test accuracy − original average test accuracy) in Fig. 5. The figure shows that this technique to exploit the strong paraphrasing capability of ChatGPT has

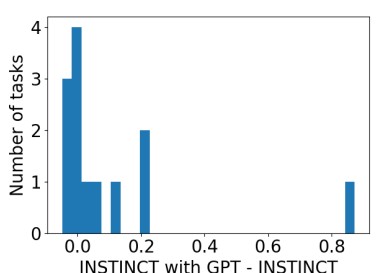

Figure 5: Improving `INSTINCT` with ChatGPT rephrasing.

Table 4: Average test accuracy achieved by (i) INSTINCT, (ii) test-time-only one-shot INSTINCT, and (iii) one-shot INSTINCT. The results including all tasks are given in Table 5 (App. D.2).

| Task | INSTINCT | test-time-only one-shot INSTINCT | one-shot INSTINCT |
|------|----------|----------------------------------|-------------------|
| antonyms | 0.8467(0.0027) | 0.8533(0.0027) | **0.8633(0.0072)** |
| auto_categorization | 0.2500(0.0330) | **0.3000(0.0125)** | **0.3000(0.0216)** |
| auto_debugging | 0.2917(0.0340) | 0.4583(0.0680) | **0.6250(0.0000)** |
| cause_and_effect | 0.5867(0.0871) | 0.6267(0.0871) | **0.7733(0.0109)** |
| common_concept | 0.2129(0.0019) | **0.2496(0.0019)** | 0.1812(0.0243) |
| diff | **1.0000(0.0000)** | **1.0000(0.0000)** | **1.0000(0.0000)** |
| informal_to_formal | **0.5534(0.0000)** | 0.5159(0.0000) | 0.5362(0.0139) |
| letters_list | **1.0000(0.0000)** | **1.0000(0.0000)** | **1.0000(0.0000)** |
| negation | 0.8167(0.0027) | **0.8567(0.0054)** | 0.8433(0.0223) |
| object_counting | 0.3400(0.0698) | 0.3567(0.0119) | **0.4600(0.0216)** |
| odd_one_out | **0.7000(0.0163)** | 0.6333(0.0109) | 0.6667(0.0054) |
| orthography_starts_with | 0.6667(0.0272) | 0.6667(0.0191) | **0.7167(0.0027)** |
| rhymes | **1.0000(0.0000)** | 0.7467(0.2068) | **1.0000(0.0000)** |
| second_word_letter | 0.1000(0.0411) | 0.2433(0.0530) | **0.4567(0.0191)** |
| sentence_similarity | 0.1400(0.0047) | 0.1600(0.0000) | **0.2400(0.0573)** |
| sum | **1.0000(0.0000)** | **1.0000(0.0000)** | 0.9933(0.0054) |
| synonyms | 0.3067(0.0491) | 0.3700(0.0694) | **0.4600(0.0047)** |
| taxonomy_animal | 0.8567(0.0599) | 0.8967(0.0495) | **0.9233(0.0098)** |
| word_sorting | 0.5133(0.0027) | **0.6200(0.0047)** | **0.6200(0.0340)** |
| word_unscrambling | **0.6333(0.0072)** | 0.5833(0.0098) | 0.5467(0.0191) |
| # best-performing tasks | 7 | 7 | **14** |
| average rank | 2.2 | 1.8 | **1.45** |

the potential to further enhance our INSTINCT algorithm (at the expense of an additional query to ChatGPT in every iteration). More details on the experiments here are given in App. D.3.

## 6 RELATED WORK

**Instruction Optimization for Black-Box LLMs.** The methods BBT (Sun et al., 2022b), BBTv2 (Sun et al., 2022a) and clip-tuning (Chai et al., 2022) have proposed to use evolutionary algorithms (EAs) to optimize the prompts for black-box LLMs. However, these methods are inapplicable to our setting because they additionally require access to the input token embeddings and output logits of the black-box LLMs, whereas the black-box LLMs we consider only allow query access. GRIPS (Prasad et al., 2022) and APO (Pryzant et al., 2023) have used edit-based operations to propose candidate instructions and performed instruction optimization in a gradient-free manner. Diao et al. (2023) have applied reinforcement learning for prompt optimization, and Guo et al. (2023) (concurrent to our paper) have adopted EAs while using an LLM as the evolutionary operator. The recent work of Zhou et al. (2023) has proposed APE, which searches for high-scoring instructions by adopting an LLM to produce candidate instructions and then using iterative re-sampling to generate other candidates similar to the promising instructions. However, these methods above are usually query-inefficient, mostly because they are based on local search and hence cannot effectively handle the exploration-exploitation trade-off (Sec. 1). Another concurrent work (Yang et al., 2023) has used an LLM as an optimizer to solve generic optimization problems and applied their method to instruction optimization. The previous method most closely related to our paper is InstructZero (Chen et al., 2023b), which has applied the query-efficient BO algorithm to instruction optimization for black-box LLMs (see Sec. 2.1). We also discuss the related works on instruction optimization for white-box LLMs (App. B.1), as well as BO and neural bandits (App. B.3). Moreover, we give a visual summarization of the related works on instruction optimization in App. B.2.

## 7 CONCLUSION

We introduce our INSTINCT algorithm to optimize the instructions for black-box LLMs. Our INSTINCT replaces the GP surrogate in BO by an NN surrogate, and couples the NN surrogate with the hidden representation learned by a pre-trained transformer. We optimize the instructions for ChatGPT, and use extensive experiments to show that our INSTINCT consistently outperforms existing methods in various tasks. A potential limitation of our INSTINCT is that it needs a numeric score during optimization and hence requires a validation set. Although we have followed the common practice from previous works, a validation set may not be easy to obtain in some applications, which will require additional techniques to attain a reliable score to guide our optimization process.

## REPRODUCIBILITY STATEMENT

We have described in Sec. 4 and App. C our detailed experimental settings, including the datasets we have adopted, data pre-processing steps, evaluation metrics, white-box and black-box LLMs adopted, values for the hyperparameters, prompting templates for different experiments, among others. In every experiment, different algorithms have used the same random seeds to ensure fair comparison. We have also submitted the code for better reproducibility.

## ACKNOWLEDGEMENTS

This research/project is supported by the National Research Foundation Singapore and DSO National Laboratories under the AI Singapore Programme (AISG Award No: AISG2-RP-2020-018).

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

## A    ETHICAL CONSIDERATIONS

Since our proposed method aims to improve the performance of LLMs (via instruction optimization) for different tasks, there may exist some ethical implications related to the usage of LLMs. Specifically, in certain maliciously designed tasks, our method may be used by a malicious party to produce harmful/inappropriate instructions. This is because currently our method only aims to maximize the test performance of the black-box LLM when selecting the instructions for any given task. Therefore, to account for such potential ethical issues and prevent the generation of harmful instructions, we may explore extensions of our method which can also account for additional objectives or constraints (e.g., harmfulness) during the optimization process.

## B    ADDITIONAL RELATED WORK

### B.1    RELATED WORKS ON INSTRUCTION OPTIMIZATION FOR WHITE-BOX LLMS

AutoPrompt (Shin et al., 2020) and FluentPrompt (Shi et al., 2022) have adopted gradient-based methods to search for an optimal sequence of discrete tokens that form an instruction. To sidestep this combinatorial optimization problem, several works (Lester et al., 2021; Li & Liang, 2021; Zhong et al., 2021) have used gradient descent to optimize a sequence of continuous task-specific vectors (i.e., soft prompt) prepended to the input prompt. RLPrompt (Deng et al., 2022) has instead trained a task-specific network inserted into a frozen pre-trained LLM via reinforcement learning (RL) reward signals. However, these methods cannot be used to optimize prompts for black-box LLMs, which are typically more powerful.

### B.2    SUMMARIZATION OF RELATED WORK ON INSTRUCTION OPTIMIZATION

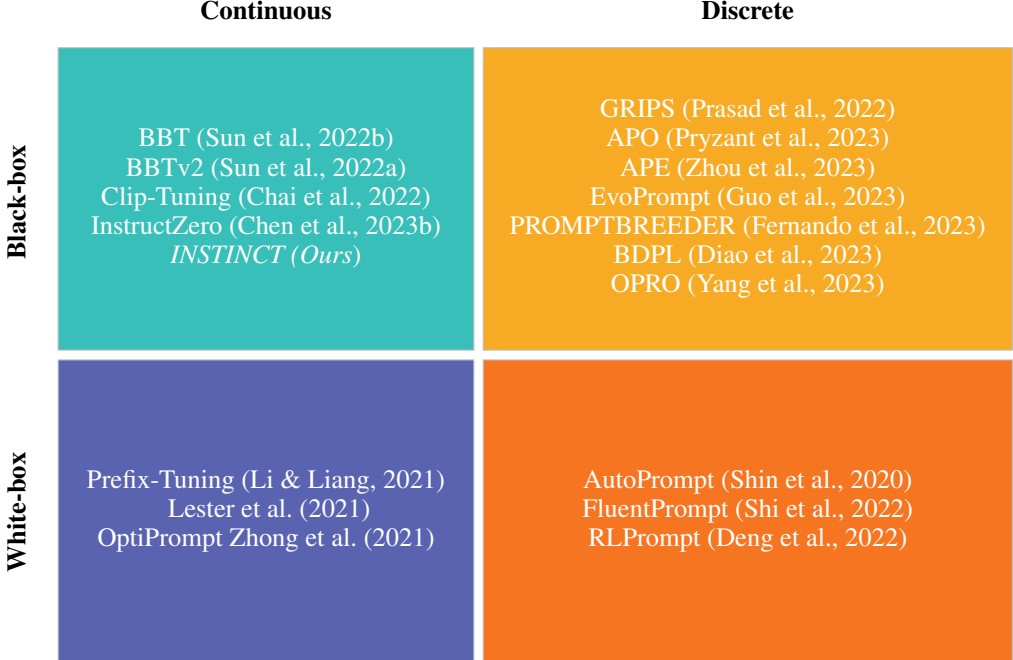

### B.3    RELATED WORK ON BAYESIAN OPTIMIZATION AND NEURAL BANDITS

Multi-armed bandits algorithms have been extensively studied to address sequential decision-marking problems by balancing the trade-off between exploration and exploitation (Lattimore & Szepesvári, 2020). Bayesian optimization (BO), also known as kernelized bandits, is a type of bandit algorithm which uses a Gaussian process (GP) (Rasmussen & Williams, 2006) to model the objective function (i.e., the reward function in bandits) (Garnett, 2023). However, traditional BO algorithms Bergstra et al. (2011); Tiao et al. (2021); Garnett (2023) can not handle the high-dimensional input objective

functions. Fortunately, leveraging the expressive power of neural networks (NNs), recent works such as NeuralUCB (Zhou et al., 2020) and NeuralTS (Zhang et al., 2021) have been proposed to better model the reward function in bandits using NNs and hence are able to deal with high-dimensional input objective functions while preserving strong theoretical guarantees. The application of neural bandits has been further extended to batched (Gu et al., 2021), federated (Dai et al., 2023) and graph-structured (Kassraie et al., 2022) settings, among others.

## C  ADDITIONAL EXPERIMENTAL DETAILS AND RESULTS

### C.1  DATASETS AND IMPLEMENTATION DETAILS

We provide comprehensive comparisons between our method and the existing baselines using various widely-used datasets. All experiments were carried out on a server with AMD EPYC processors and NVIDIA A100 GPUs.

**Prompting Templates.** Carefully designed language input prompts are important to elicit expected responses from the LLMs. We follow InstructZero for the template designs.

For instruction generation, we use the following prompting template with five-shot demonstrations (Fig. 6) where each [INPUT] and [OUTPUT] pairs are replaced with the input-output pairs from the exemplar set $E = \{(x_\tau, y_\tau)\}_{\tau=1}^{\kappa}$ when $\kappa = 5$. The output from the LLM is our instruction $\rho$.

**Instruction Generation Template**

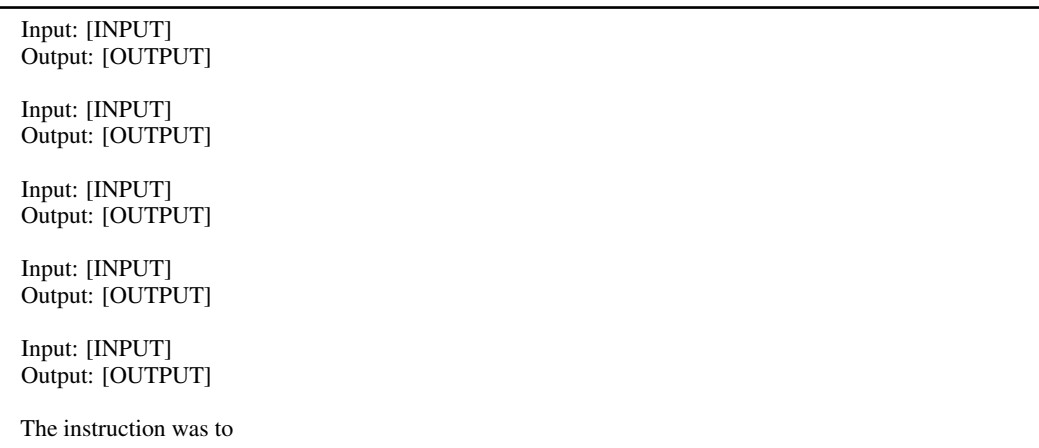

Figure 6: The prompt for our white-box LLM to generate instructions.

To evaluate an instruction, we use the following prompting template on a test input (Fig. 7) where [INSTRUCTION] is replaced with the instruction $\rho$ and [TEST INPUT] is replaced with test input from a separate test set $D_T$.

**Evaluation Template**

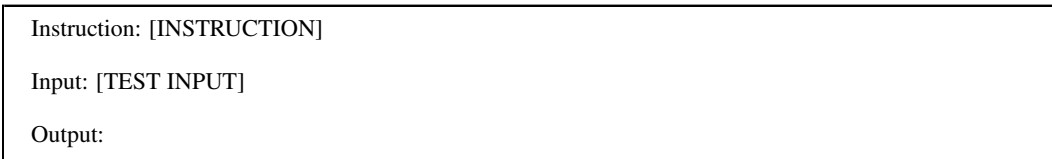

Figure 7: The prompt for the black-box LLM to generate answer/output.

**Datasets & Preprocessing.** The 30 datasets for instruction induction in Sec. 4.1 are the same as those in InstructZero (Chen et al., 2023b). We omitted 2 datasets, namely CS Algorithms, and ASCII, because the test datasets for these two tasks are not open-sourced. For the SAMSum dataset, we

select 200 data points from the original test dataset as a test dataset to save the cost of evaluation (i.e., the cost of calling ChatGPT API). For the arithmetic reasoning datasets (i.e., GSM8K, AQUARAT, SVAMP), we process the dataset the same way as APE (Zhou et al., 2023) does. For GSM8K and AQUARAT, we use all test data from each corresponding test dataset to evaluate the test accuracy of the instruction. For AQUARAT, we sample 400 data points from its test datasets as $D_T$ for evaluating the test accuracy to save the cost of the evaluation. For all arithmetic reasoning datasets, we sample 200 data points from each corresponding training dataset as the validation dataset $D_V$.

**Evaluation Metrics.** For instruction induction, we use the F1 score for "common_concept", "informal_to_formal" and SAMSum; we use the exact set matching for "orthography_starts_with" and "taxonomy_animal"; we check whether the output label is contained in the model output for the task of "synonyms"; and we adopt the "exact match" metric for the rest of instruction induction tasks. For the SAMSum dataset, we additionally provide ROUGE-1, ROUGE-2, and ROUGE-L (Lin, 2004) as the evaluation metrics. For the arithmetic reasoning datasets, we use the same way as APE (Zhou et al., 2023) to extract the answer (e.g., numbers or choices) from the generated text and use accuracy as the metric.

**White/Black-box Models.** We follow InstructZero to use Vicuna-13B as the default white-box model for instruction generation and GPT-3.5-turbo as the default black-box model for evaluation. However, the versions for these models are not specified by the InstructZero paper. Especially for GPT-3.5-turbo, the model is continually updated by OpenAI. To ensure fair comparison and reproducibility, we use GPT-3.5-turbo-0301 (which will be supported by OpenAI until at least June 2024) and Vicuna-13B-v1.1 as the default model choices for all the experiments carried out in this paper.

**Hyperparameter Details for the Neural Bandit Algorithm.** We set $\lambda = 0.1$ (Sec. 3.1) and $\nu_t = 1$ (Sec. 3.2) in all experiments. When doing the random projection, the elements in the projection matrix are sampled i.i.d. from $Uni(-1, 1)$. We choose the number of hidden representations in the discrete domain $\widetilde{Z}$ to be $10000$ and at each iteration of our INSTINCT, we randomly sample $1000$ data points from the $\widetilde{Z}$ to evaluate the NeuralUCB acquisition function to accelerate our algorithm. We stack an MLP on top of the hidden representations of the pre-trained transformer language model. The MLP has an input dimension of $5120$, an output dimension of $1$, and a hidden layer of size $100$. We train the MLP following to minimize the mean squared error (MSE) loss for $1000$ iterations after each new observation point. A default learning rate of $0.001$ is used.

**Repeated Experiments with Seeding.** For both our INSTINCT and InstructZero, when we perform a grid search over the intrinsic dimension (within $\{10, 50, 100\}$) and the number of soft tokens (within $\{3, 5, 10\}$) using the validation set for each task, we only conduct this grid search for 1 of the 3 trials, and directly use the best parameters found in the first trial in the remaining 2 trials. This is done to save computational costs.

## C.2 INSTRUCTION INDUCTION

For better distinguishability, we only presented the tasks for which any method has an average test accuracy less than $0.8$ (i.e., more challenging tasks) in the main text. A full comparison including all tasks is given in Table 9 below. Overall, our INSTINCT significantly outperforms the APE and InstructZero baselines, achieving the best performance in 19 out of the 30 instruction induction tasks. To assess the performance of the methods from their ranks in each task, INSTINCT also achieves the highest average ranking of $1.53$ over all instruction induction tasks.

**Visualizing the Optimization Process.** We additionally investigate the optimization process of our INSTINCT algorithm in finding the best instruction. Fig. 8 presents the average test performance (over 3 random seeds) of the best instruction so far over optimization iterations. We partition the optimization process into three stages, namely stage A, B, C, which corresponds to iteration 30, 90, 150, respectively. We see steady improvements as the optimization progresses, producing better instructions that capture the input-output relationships of the tasks. The instructions at stage C matches closely to the task descriptions provided in Fig. 8. Interestingly, we also observe that the instruction that performs best on LLM does not necessarily correspond to human perception. For example, in the task of *object_counting*, the instruction outputted at stage B may seem more appropriate to a human being as compared to the instruction at stage C. However, the latter achieves better test performance. This further necessitates an automatic instruction tuning algorithm like

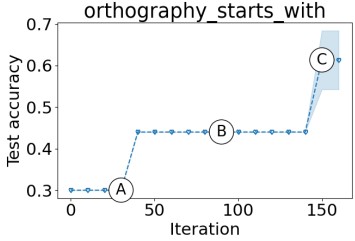

Task description: given a sentence and a letter, output the words that start with the letter in the sentence.

| Iteration | Instruction |
|-----------|-------------|
| A | The instruction was to find a word that could be formed by rearranging the letters of the given word |
| B | The instruction was to find the word that the input corresponds to, and output it |
| C | The instruction was to output the word that starts with the letter that was inputted |

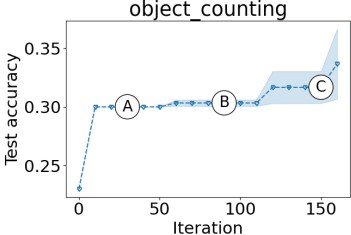

Task description: given a sentence, output the number of objects in the sentence.

| Iteration | Instruction |
|-----------|-------------|
| A | The instruction was to output the number of items that the speaker has, given the list of items that the speaker possesses |
| B | The instruction was to output the number of objects mentioned in the input |
| C | The instruction was to output the number of items the player has, but the player has entered the number of items instead |

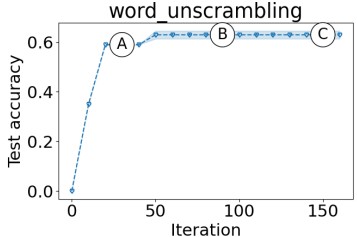

Task description: given a list of shuffled letters, rearrange the letters to form a meaningful word.

| Iteration | Instruction |
|-----------|-------------|
| A | The instruction was to output the word that the input word spells when the letters are rearranged in a specific order |
| B | The instruction was to output the word that is formed by rearranging the letters of the given word |
| C | The instruction was to output the word that is formed by rearranging the letters of the given word |

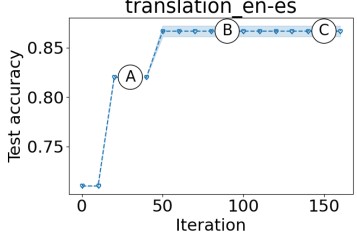

Task description: translate the words from English to Spanish.

| Iteration | Instruction |
|-----------|-------------|
| A | The instruction was to translate the words from Spanish to English |
| B | The instruction was to translate the words from English to Spanish |
| C | The instruction was to translate the words from English to Spanish |

Figure 8: The test accuracy of the best instruction found as the iteration increases. The quality of the instruction (i.e., test accuracy) increases as more iterations are used to perform our `INSTINCT`.

`INSTINCT` which streamlines the process to generate task-specific instructions that work the best on a given black-box LLM.

## C.3 IMPROVING ZERO-SHOT CHAIN-OF-THOUGHT INSTRUCTIONS

To improve the chain-of-thought instruction for arithmetic reasoning tasks, we use the following instruction generation prompt for the white-box LLM. Note that when finding the chain-of-thought instruction, we fix the intrinsic dimension $d'$ to be 1000 and search the number of soft tokens $N_z$ over $\{3, 5, 10\}$ for both InstructZero and `INSTINCT`. The intrinsic dimension is set to be higher because we find that when the intrinsic dimension is smaller than 1000, the variety of the generated instructions will be small (i.e., different soft prompts will result in the same instruction). Increasing the number of intrinsic dimensions will increase the L2 norm of the soft prompt (as we will investigate in App. E.2) and thus affect the generated instruction more (i.e., generating different instructions given different soft prompts). As shown in Fig. 9, we ask the white-box LLM to generate prompts

to solve the math problems and we provide 3 example chain-of-thought instructions to guide the white-box LLM to generate chain-of-thought style instructions as similarly used by (Yang et al., 2023). These examples are important since the white-box LLM needs to know what are the possible instructions for other LLMs to do chain-of-thought. The other part of the setting is the same as the setting in the instruction induction task.

**Instruction Generation Template for Chain-of-thought**

I have some instruction examples for solving school math problems.

Instruction:
Let's figure it out!

Instruction:
Let's solve the problem.

Instruction:
Let's think step by step.

Write your new instruction that is different from the examples to solve the school math problems.

Instruction:

Figure 9: The prompt for our white-box LLM to generate chain-of-thought instructions.

# D  MORE DETAILS ON ABLATION STUDY

## D.1  HIDDEN REPRESENTATIONS GIVE BETTER SIMILARITY MEASURE (MORE DETAILS)

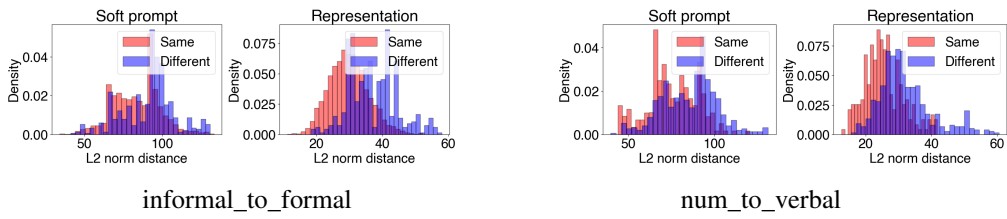

Figure 10: Two additional tasks for Fig. 4. See the caption of Fig. 4 for a detailed explanation.

Here we give more discussions as to why the hidden representations lead to better similarity measures compared with the original soft prompts, which has been verified by our experiments in Sec. 5.

Note that we feed every soft prompt $z_t$ (i.e., a *continuous vector*) to the white-box LLM $w$ to produce the instruction $\rho_t$ (i.e., a sentence), which is then passed to the black-box LLM $f$ for evaluation (Fig. 2). As a result, there exist multiple soft prompts (i.e., continuous vectors) that map to the same instruction. Therefore, it is important for an instruction optimization algorithm to take this into account in order to achieve efficient exploration of the space of soft prompts (i.e., to avoid unnecessary queries at multiple soft prompts which map to the same instruction). Despite their instruction-coupled kernel (Chen et al., 2023b), InstructZero is *unable to account for this issue* when computing the similarity between an already-queried soft prompt and a new soft prompt, because its similarity measure reduces to the standard L2 distance-based kernel (i.e., the Matérn kernel). In contrast, our `INSTINCT` algorithm is able to take this issue into account because it builds the NN surrogate on top of *the hidden representations of the soft prompts* (Sec. 3.1) which makes it easier to identify pairs of soft prompts that map to the same instruction.

Fig. 10 below presents the results for two additional tasks (in addition to the two tasks shown in the main paper in Fig. 4). The results from these additional figures lead to the same interpretations as Fig. 4 in the main paper, i.e., for pairs of soft prompts leading to the same instruction, the hidden representations make it much easier to identify that they should be assigned high similarity values.

Table 5: Table containing the results for all tasks in our ablation study on further improving our `INSTINCT` algorithm via one-shot in-context-learning (Sec. 5) The results here correspond to Table 4 in Sec. 5 in the main paper.

| Task | zero-shot `INSTINCT` | test-time-only one-shot `INSTINCT` | one-shot `INSTINCT` |
|---|---|---|---|
| active_to_passive | 0.9700(0.0245) | **1.0000(0.0000)** | 0.9967(0.0027) |
| antonyms | 0.8467(0.0027) | 0.8533(0.0027) | **0.8633(0.0072)** |
| auto_categorization | 0.2500(0.0330) | **0.3000(0.0125)** | **0.3000(0.0216)** |
| auto_debugging | 0.2917(0.0340) | 0.4583(0.0680) | **0.6250(0.0000)** |
| cause_and_effect | 0.5867(0.0871) | 0.6267(0.0871) | **0.7733(0.0109)** |
| common_concept | 0.2129(0.0019) | **0.2496(0.0019)** | 0.1812(0.0243) |
| diff | **1.0000(0.0000)** | **1.0000(0.0000)** | **1.0000(0.0000)** |
| first_word_letter | 0.9300(0.0531) | **1.0000(0.0000)** | **1.0000(0.0000)** |
| informal_to_formal | **0.5534(0.0000)** | 0.5159(0.0000) | 0.5362(0.0139) |
| larger_animal | **0.9367(0.0027)** | 0.9267(0.0027) | 0.9300(0.0000) |
| letters_list | **1.0000(0.0000)** | **1.0000(0.0000)** | **1.0000(0.0000)** |
| negation | 0.8167(0.0027) | **0.8567(0.0054)** | 0.8433(0.0223) |
| num_to_verbal | **1.0000(0.0000)** | **1.0000(0.0000)** | **1.0000(0.0000)** |
| object_counting | 0.3400(0.0698) | 0.3567(0.0119) | **0.4600(0.0216)** |
| odd_one_out | **0.7000(0.0163)** | 0.6333(0.0109) | 0.6667(0.0054) |
| orthography_starts_with | 0.6667(0.0272) | 0.6667(0.0191) | **0.7167(0.0027)** |
| periodic_elements | 0.9267(0.0272) | 0.9800(0.0000) | **1.0000(0.0000)** |
| rhymes | **1.0000(0.0000)** | 0.7467(0.2068) | **1.0000(0.0000)** |
| second_word_letter | 0.1000(0.0411) | 0.2433(0.0530) | **0.4567(0.0191)** |
| sentence_similarity | 0.1400(0.0047) | 0.1600(0.0000) | **0.2400(0.0573)** |
| sentiment | 0.8967(0.0144) | **0.9167(0.0072)** | 0.8933(0.0027) |
| singular_to_plural | **1.0000(0.0000)** | 0.9933(0.0027) | 0.8433(0.0650) |
| sum | **1.0000(0.0000)** | **1.0000(0.0000)** | 0.9933(0.0054) |
| synonyms | 0.3067(0.0491) | 0.3700(0.0694) | **0.4600(0.0047)** |
| taxonomy_animal | 0.8567(0.0599) | 0.8967(0.0495) | **0.9233(0.0098)** |
| translation_en-de | **0.8400(0.0047)** | 0.8300(0.0082) | 0.8367(0.0027) |
| translation_en-es | 0.8800(0.0000) | 0.8700(0.0047) | **0.8833(0.0027)** |
| translation_en-fr | 0.8300(0.0205) | 0.8767(0.0072) | **0.8867(0.0119)** |
| word_sorting | 0.5133(0.0027) | **0.6200(0.0047)** | **0.6200(0.0340)** |
| word_unscrambling | **0.6333(0.0072)** | 0.5833(0.0098) | 0.5467(0.0191) |
| # best-performing tasks | 11 | 11 | **19** |
| average rank | 2.13 | 1.83 | **1.53** |

## D.2 IMPROVING `INSTINCT` VIA ONE-SHOT IN-CONTEXT LEARNING

We have demonstrated in Table 4 (in the main text) the improved performance of instruction induction when incorporating one-shot in-context learning into our algorithm. Table 5 shows the results of all 30 instruction induction tasks. One-shot `INSTINCT` is the best-performing method in 19 out of 30 tasks with a highest average ranking of 1.53. Therefore, we draw the same conclusion as the main text that our `INSTINCT` is compatible with in-context learning and has wider potential applications of our INSTINCT through its combination with in-context learning.

Now, we elaborate on the necessary modifications in the implementation to carry out one-shot in-context learning with `INSTINCT`. To directly adopt in-context learning at test time (i.e., for our test-time-only one-shot `INSTINCT` algorithm), we modify the evaluation template to include one exemplar as a demonstration. Referring to Fig. 11, [INSTRUCTION] is replaced with the instruction $\rho$, [INPUT] and [OUTPUT] pairs are replaced with one exemplar's input and output, and [TEST

INPUT] is replaced with test input from a separate test set $D_T$. For one-shot INSTINCT, we use the one-shot evaluation template (Fig. 11) for both validation and test.

**One-shot Evaluation Template**

Instruction: [INSTRUCTION]

Input: [INPUT]
Output: [OUTPUT]

Input: [TEST INPUT]

Output:

Figure 11: The prompt for the black-box LLM to generate answer/output.

### D.3 IMPROVE INSTINCT WITH CHATGPT REPHRASING (MORE DETAILS)

To rephrase the instruction using ChatGPT to further improve the performance of INSTINCT as discussed in Sec. 5, we use the prompting template as shown in Fig. 12 to rephrase every instruction $\rho_t$ generated by Vicuna in every iteration $t$. The [INSTRUCTION] in Fig. 12 is replaced by the instruction $\rho_t$ generated by Vicuna and the [INPUT] and [OUTPUT] are the same exemplars we used for generating the instruction from Vicuna (i.e., the set $E$ of exemplars in Fig. 2).

**Rephrasing Template**

We have an instruction to write an output for each input: [INSTRUCTION]. Here are some input-output pairs:
Input: [INPUT]
Output: [OUTPUT]

Input: [INPUT]
Output: [OUTPUT]

Input: [INPUT]
Output: [OUTPUT]

Input: [INPUT]
Output: [OUTPUT]

Input: [INPUT]
Output: [OUTPUT]

We need a better rephrasing of the instruction to guide a person in writing a correct output for an input. The rephrased instruction is to

Figure 12: The prompt template for rephrasing instruction generated by INSTINCT using ChatGPT to further boost the performance.

We also present Table. 6 containing results corresponding to Fig. 5 in the main text. We observe the potential to further improve difficult tasks by exploiting the strong paraphrasing capability of ChatGPT.

### D.4 EFFECTIVENESS OF PRINCIPLED EXPLORATION

We verify the effectiveness of the principled exploration of our INSTINCT algorithm facilitated by the uncertainty term $\sigma(\cdot, \cdot)$ in equation 2, which is one of the important strengths of our

Table 6: Test accuracy achieved by our technique of ChatGPT rephrasing (Sec. 5) to further improve the performance of our INSTINCT. The results here correspond to Fig. 5 in the main paper.

|  | INSTINCT | INSTINCT+ChatGPT |
|---|---|---|
| auto_categorization | 0.2500(0.0330) | **0.2800(0.0027)** |
| auto_debugging | 0.2917(0.0340) | **0.2917(0.0113)** |
| cause_and_effect | 0.5867(0.0871) | **0.7867(0.0960)** |
| common_concept | **0.2129(0.0019)** | 0.1644(0.0024) |
| informal_to_formal | **0.5534(0.0000)** | 0.5533(0.0057) |
| odd_one_out | 0.7000(0.0163) | **0.7067(0.0048)** |
| object_counting | 0.3400(0.0698) | **0.5400(0.0072)** |
| orthography_starts_with | 0.6667(0.0272) | **0.7133(0.0048)** |
| second_word_letter | 0.1000(0.0411) | **0.9733(0.0059)** |
| sentence_similarity | **0.1400(0.0047)** | 0.1367(0.0524) |
| synonyms | **0.3067(0.0491)** | 0.2867(0.0398) |
| word_sorting | 0.5133(0.0027) | **0.6200(0.0047)** |
| word_unscrambling | **0.6333(0.0072)** | 0.6067(0.0119) |

INSTINCT (Sec. 3.4). To achieve this, we compare the performance of our INSTINCT algorithm when the weighting parameter $\nu_t$ (equation 2) is set to (i) the default value of $\nu_t = 1$ (i.e., with exploration) which is used in all our experiments and (ii) $\nu_t = 0$ (i.e., no exploration). As shown in Table 7, having principled exploration helps dramatically improve the instruction induction performance, which verifies the necessity of the principled exploration in our INSTINCT algorithm.

Table 7: Performance of INSTINCT with $\nu_t = 1$ (with our principled exploration) and $\nu_t = 0$ (no exploration). See a detailed explanation in App. D.4.

| Algorithm | $\nu_t = 1$ | $\nu_t = 0$ |
|---|---|---|
| active_to_passive | **1.000000** | 0.950000 |
| antonyms | 0.850000 | **0.860000** |
| auto_categorization | 0.300000 | **0.320000** |
| common_concept | **0.217614** | 0.210471 |
| informal_to_formal | **0.553387** | 0.553387 |
| negation | **0.820000** | 0.810000 |
| object_counting | **0.410000** | 0.380000 |
| odd_one_out | **0.740000** | 0.680000 |
| second_word_letter | **0.160000** | 0.100000 |
| sentence_similarity | **0.150000** | 0.140000 |
| sentiment | **0.930000** | 0.880000 |
| synonyms | **0.390000** | 0.370000 |
| taxonomy_animal | 0.930000 | **0.950000** |
| translation_en-fr | **0.880000** | 0.820000 |
| word_sorting | 0.510000 | **0.520000** |
| word_unscrambling | **0.650000** | 0.410000 |

# E  MORE TECHNICAL DETAILS ON OUR INSTINCT ALGORITHM (SEC. 3)

## E.1  MORE TECHNICAL DETAILS ON OUR PRINCIPLED UNCERTAINTY MEASURE

Here we explain the detailed calculation of our principled measure of uncertainty $\sigma_{t-1}(g(z); \theta_{t-1})$ Sec. 3.2), which has been derived based on the theory of neural tangent kernel (NTK) (Jacot et al., 2018; Arora et al., 2019) and neural bandits (Zhou et al., 2020; Zhang et al., 2021).

We use $\nabla_\theta m(g(z), \theta_{t-1})$ to represent the gradient of the NN parameters $\theta$ evaluated at $\theta_{t-1}$, denote by $p$ the total number of parameters of the NN surrogate, and use $I_{p \times p}$ to represent the $p \times p$-dimensional identity matrix. In iteration $t$, i.e., after the first $t - 1$ observations $\{(g(z_\tau), h_\tau)\}_{\tau=1}^{t-1}$

have been collected, we firstly calculate the following $p \times p$-dimensional matrix:

$$V_{t-1} = \sum_{\tau=1}^{t-1} \nabla_\theta m(g(z), \theta_{t-1}) \nabla_\theta m(g(z), \theta_{t-1})^\top + \lambda I_{p \times p}. \tag{3}$$

Next, our uncertainty about the function value $h(z)$ at $z$ is calculated as:

$$\sigma_{t-1}(g(z); \theta_{t-1}) = \sqrt{\nabla_\theta m(g(z), \theta_{t-1})^\top V_{t-1}^{-1} \nabla_\theta m(g(z), \theta_{t-1})}. \tag{4}$$

Formally, $\sigma_{t-1}(g(z); \theta_{t-1})$ is the *Gaussian process (GP) posterior standard deviation* (Rasmussen & Williams, 2006) when the empirical NTK is used as the kernel: $k(z_1, z_2) = \nabla_\theta m(g(z_1), \theta_{t-1})^\top \nabla_\theta m(g(z_2), \theta_{t-1})$. Note that when calculating the uncertainty $\sigma_{t-1}(g(z); \theta_{t-1})$ here, we have treated the hidden representation $g(z)$ as the input, which is justified because the hidden representation is fixed since we freeze the parameters of the white-box LLM $w$ (i.e., the pre-trained transformer).

The calculation of $\sigma_{t-1}(g(z); \theta_{t-1})$ in equation 4 requires inverting the matrix $V_{t-1}$ which is usually computationally prohibitive since the number $p$ of parameters of the NN is usually very large. Therefore, we have followed the common practice in neural bandits (e.g., adopted by both NeuralUCB (Zhou et al., 2020) and NeuralTS (Zhang et al., 2021)) to use a diagonal approximation of $V_{t-1}$. That is, we only keep the diagonal elements of $V_{t-1}$ and set all other matrix elements to 0, which allows us to sidestep the computational cost of matrix inversion.

### E.2    DETAILED EXPLANATION ON THE IMPACT OF THE INTRINSIC DIMENSION

In this section, through both theoretical analysis and empirical demonstrations, we analyze the impact of the intrinsic dimension $d'$ of the random variable drawn from the Sobel sequence on the norm of the soft prompt. Our results here provide justifications for our discussion in Sec. 3.2 in the paragraph **Generating the Discrete Domain $\widetilde{Z}$**.

Let $z' \in \mathbb{R}^{d'}$ be the vector drawn from the Sobel sequence where $d'$ is the intrinsic dimension. Note that each element of $z'$ is identically and independently distributed (i.i.d.), i.e., $z'_j \sim Uni(0, 1)$. We use $A \in \mathbb{R}^{d \times d'}$ to denote a random projection matrix where each element $A_{ij} \sim Uni(-1, 1)$ is also i.i.d. distributed.

Now, consider $z = Az'$ and we want to calculate $\|z\|^2$. We can alternatively express

$$A = \begin{bmatrix} A_1 \\ A_2 \\ \vdots \\ A_d \end{bmatrix}, \quad Az' = \begin{bmatrix} A_1 z' \\ A_2 z' \\ \vdots \\ A_d z' \end{bmatrix}$$

where each $A_i$ is a row vector of dimension $d'$ and $A_i z' = \sum_{j=1}^{d'} A_{ij} z'_j$.

Then,

$$\mathbb{E}\left[\|z\|^2\right] = \mathbb{E}\left[\|Az'\|^2\right]$$

$$= \mathbb{E}\left[\sum_{i=1}^{d} (A_i z')^2\right]$$

$$= \mathbb{E}\left[\sum_{i=1}^{d} \left(\sum_{j=1}^{d'} A_{ij} z'_j\right)^2\right]$$

$$\overset{(a)}{=} \sum_{i=1}^{d} \sum_{j=1}^{d'} \mathbb{E}\left[\left(A_{ij} z'_j\right)^2\right]$$

$$\overset{(b)}{=} dd' \mathbb{E}\left[A_{ij}^2\right] \mathbb{E}\left[z'^2_j\right]$$

where (a) follows from the fact that all elements are independent random variables and (b) follows from i.i.d. $A_{ij}$ and i.i.d. $z'_j$. Therefore, we have shown that $\mathbb{E}\left[\|z\|^2\right] \propto d'$.

We perform a synthetic experiment to verify the result we derived above. Specifically, for a fixed intrinsic dimension, we sample 1000 Sobol sequences and use random projection to project them to the soft prompt space. We compute the average square of the L2 norm of the 1000 soft prompts. We vary the intrinsic dimension from 10 to 1000. As shown in Fig. 13, when increasing the number of intrinsic dimensions, the square of the L2 norm of the corresponding soft prompts increases linearly with the number of intrinsic dimensions.

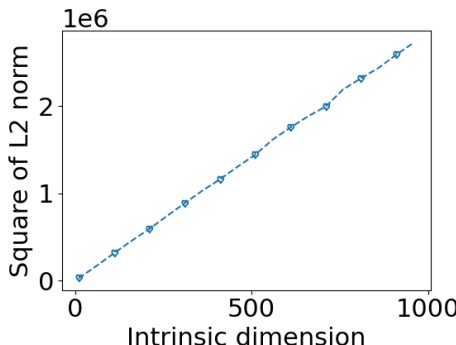

Figure 13: The square of the L2 norm of the soft prompt as the number of intrinsic dimensions increases from 10 to 1000.

Table 8: The best instruction discovered by our INSTINCT algorithm for every instruction induction task (Sec. 4.1).

| Task | Best instruction |
| --- | --- |
| active_to_passive | The instruction was to flip the subject and verb in each sentence, but keep the preposition the same |
| antonyms | The instruction was to take a word and change it to its opposite |
| auto_categorization | The instruction was to create a list of things that the input could be associated with, and the output would be the category that the input belongs to |
| auto_debugging | The instruction was to add a line of code to the input to change the output |
| cause_and_effect | The instruction was to identify the sentence that is the cause of the effect in the input sentence pair |
| common_concept | The instruction was to "involve" the objects mentioned in the input, so the answer would be "involve oscillations" for the input "guitars, pendulums" |
| diff | The instruction was to be:. Input: 41 13. Output: 28. The instruction was to be:. Input: 72 31. Output: 41. The instruction was to be:. Input: 125 35. Output: 90. |
| first_word_letter | The instruction was to input the word "year" into the computer and the output was "y". The instruction was to input the word "trust" into the computer and the output was "t". The instruction was to input the word "qualification" into the computer and the output was "q". |
| informal_to_formal | The instruction was to convert the input sentence into an output sentence that is grammatically correct and idiomatic in English |
| larger_animal | The instruction was to create a program that takes an input of two animals and outputs the animal that is bigger |
| letters_list | The instruction was to output the input with a space after each letter |
| negation | The instruction was to make the output false by adding the word "not" to the input |
| num_to_verbal | The instruction was to convert numbers to words |
| object_counting | The instruction was to output the number of objects in the input list, but the number of objects was not limited to just whole numbers |
| odd_one_out | The instruction was to find the word that is most dissimilar to the others in the group |
| orthography_starts_with | The instruction was to output the word that starts with the letter that was inputted |
| periodic_elements | The instruction was to write a program that takes an input of a number and outputs the name of the element that corresponds to that number on the periodic table |
| rhymes | The instruction was to output the first word that appeared in the input text, with no extra spaces |
| second_word_letter | The instruction was to input the word 'year' and output the word 'e'. The input 'year' was not found in the list of words provided |
| sentence_similarity | The instruction was to classify the following sentences based on their degree of certainty, using the following scale:. 1 - probably. 2 - possibly. 3 - probably not. The results are as follows:. Sentence 1: 3 - probably not. Sentence 2: 2 - possibly. |
| sentiment | The instruction was to output positive or negative based on the sentiment of the review, regardless of whether the review was actually positive or negative |
| singular_to_plural | The instruction was to add -s to the end of each word, which indicates plural |
| sum | The instruction was to add the numbers of the two input numbers |
| synonyms | The instruction was to create a list of words that could be used in the same way as the original words |
| taxonomy_animal | The instruction was to output the name of an animal that starts with the letter "H". Input: horse, pajamas, scarf, hippo, egg, whale. Output: whale, horse, pajamas, scarf, hippo. The instruction was to output the name of an animal |
| translation_en-de | The instruction was to translate the input words into German, and the output was the German word that corresponded to the input word |
| translation_en-es | The instruction was to translate the following text:. Input: press. Output: prensa. Input: participant. Output: participante. Input: company. Output: compañía. Input: outline. Output: contorno. Input: label. Output: etiquetar. The translation is not accurate, the output does |
| translation_en-fr | The instruction was to be given in French, but the words were often mispronounced, and the. Input: student. Output: étudiant. Input: doctor. Output: médecin. Input: lawyer. Output: avocat. Input: engineer. Output: ingénieur. Input: chef. Output: chef |
| word_sorting | The instruction was to "List: Serve wittgenstein emissive explicable ahmedabad grebe" and the output was "ahmedabad emissive explicable grebe serve wittgenstein" |
| word_unscrambling | The instruction was to output the word that is formed by rearranging the letters of the given word |

Table 9: Test accuracy (standard error) for the best instruction discovered by different algorithms for all instruction induction tasks (Sec. 4.1). The results are obtained using 3 independent trials with different random seeds. The Table corresponds to Table 1 in the main paper, except that all tasks are shown here.

| Algorithm | APE | InstructZero | INSTINCT |
|---|---|---|---|
| active_to_passive | **1.0000(0.0000)** | 0.9967(0.0027) | 0.9700(0.0245) |
| antonyms | 0.6367(0.1416) | 0.8267(0.0072) | **0.8467(0.0027)** |
| auto_categorization | 0.2500(0.0094) | **0.2567(0.0119)** | 0.2500(0.0330) |
| auto_debugging | 0.2917(0.0340) | **0.3750(0.0000)** | 0.2917(0.0340) |
| cause_and_effect | 0.5733(0.0891) | **0.8133(0.0109)** | 0.5867(0.0871) |
| common_concept | 0.0691(0.0207) | 0.0864(0.0398) | **0.2129(0.0019)** |
| diff | 0.6733(0.2667) | 0.6933(0.2224) | **1.0000(0.0000)** |
| first_word_letter | **1.0000(0.0000)** | **1.0000(0.0000)** | 0.9300(0.0531) |
| informal_to_formal | **0.5736(0.0026)** | 0.5310(0.0024) | 0.5534(0.0000) |
| larger_animal | 0.8967(0.0054) | 0.9000(0.0408) | **0.9367(0.0027)** |
| letters_list | **1.0000(0.0000)** | 0.5900(0.1674) | **1.0000(0.0000)** |
| negation | 0.7533(0.0109) | 0.7767(0.0136) | **0.8167(0.0027)** |
| num_to_verbal | 0.9967(0.0027) | **1.0000(0.0000)** | **1.0000(0.0000)** |
| object_counting | **0.3633(0.0191)** | 0.3600(0.0929) | 0.3400(0.0698) |
| odd_one_out | 0.6333(0.0144) | 0.6133(0.0871) | **0.7000(0.0163)** |
| orthography_starts_with | 0.4567(0.1477) | 0.5067(0.0871) | **0.6667(0.0272)** |
| periodic_elements | **0.9267(0.0218)** | 0.8667(0.0606) | **0.9267(0.0272)** |
| rhymes | 0.1567(0.0640) | **1.0000(0.0000)** | **1.0000(0.0000)** |
| second_word_letter | **0.7467(0.2028)** | 0.4333(0.1872) | 0.1000(0.0411) |
| sentence_similarity | 0.0000(0.0000) | 0.0000(0.0000) | **0.1400(0.0047)** |
| sentiment | **0.9133(0.0144)** | 0.8767(0.0242) | 0.8967(0.0144) |
| singular_to_plural | **1.0000(0.0000)** | 0.9867(0.0109) | **1.0000(0.0000)** |
| sum | 0.6733(0.2667) | **1.0000(0.0000)** | **1.0000(0.0000)** |
| synonyms | **0.3600(0.0759)** | 0.2767(0.0925) | 0.3067(0.0491) |
| taxonomy_animal | 0.3467(0.2341) | 0.7167(0.0838) | **0.8567(0.0599)** |
| translation_en-de | **0.8400(0.0047)** | 0.8233(0.0098) | **0.8400(0.0047)** |
| translation_en-es | 0.8700(0.0000) | 0.8733(0.0054) | **0.8800(0.0000)** |
| translation_en-fr | **0.8867(0.0027)** | 0.8767(0.0027) | 0.8300(0.0205) |
| word_sorting | 0.3300(0.0374) | 0.3100(0.1143) | **0.5133(0.0027)** |
| word_unscrambling | 0.4400(0.1389) | 0.5500(0.0170) | **0.6333(0.0072)** |
| #tasks that perform the best | 12 | 7 | **19** |
| #tasks that perform the second | 5 | 14 | 6 |
| Average ranking | 2.03 | 2.07 | **1.53** |

### E.3 PERFORMANCE OF INSTRUCTZERO AND INSTINCT UNDER DIFFERENT SOFT PROMPT DIMENSIONALITIES

We vary the soft prompt dimensionality by changing the number of soft tokens to see how the performance of INSTINCT and InstructZero changes. Specifically, we vary the number of soft tokens among $[3, 5, 10]$ which corresponds to soft prompt dimensionalities of $[15360, 26500, 51200]$ respectively. Fig. 14 shows the performance comparison between INSTINCT and InstructZero. The performance is measured by the fraction of tasks that each approach achieves the best performance among the performance of InstructZero and INSTINCT. As the soft prompt dimensionality increases, the performance of InstructZero drops. When the soft prompt dimensionality is 51200, the fraction of achieving the best performance for InstructZero drops dramatically compared to when the soft prompt dimensionality is 5120. While for INSTINCT, the performance does not drop significantly as the soft prompt dimensionality increases. Fig. 15 shows the test accuracy of the instructions generated from InstructZero and INSTINCT for the detailed tasks when the soft prompt dimensionality changes. Similarly, the performance of InstructZero generally drops when the soft prompt dimensionality increases while the performance of INSTINCT remains stable and better than InstructZero.

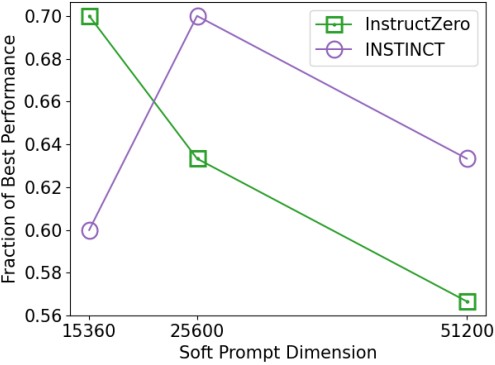

Figure 14: The fraction of tasks that one approach achieves the best performance among 30 instruction induction tasks under different soft prompt dimensionalities. Higher is better.

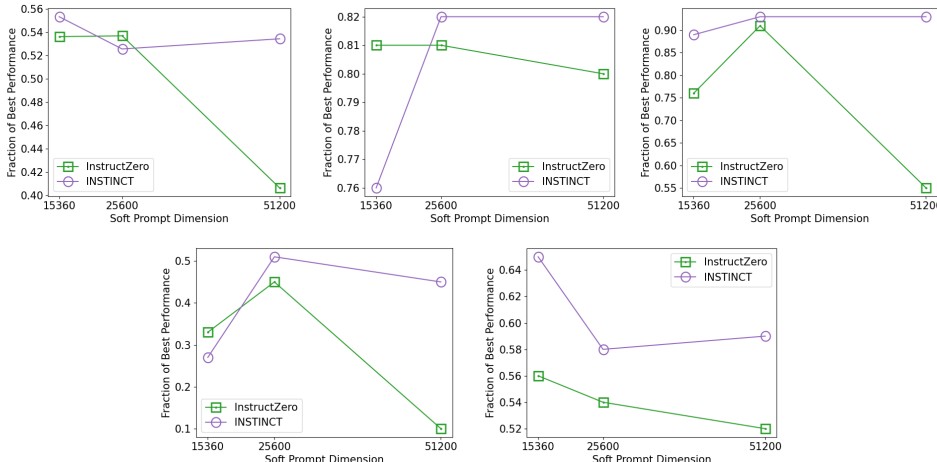

Figure 15: Test accuracy on different instruction induction tasks under different soft prompt dimensionalities.

# F    SPEEDUP OF USING PRE-COMPUTATION

We have conducted additional experiments to compare the running time of our `INSTINCT` algorithm with and without the pre-computation of the representations. The results (Table 10) show that the pre-computation dramatically reduces the running time of our `INSTINCT` algorithm.

Table 10: The running time (± standard error) of our `INSTINCT` algorithm with and without pre-computation. The results are averaged on 20 instruction induction tasks in Table 1.

| Algorithm | Running time (165 iterations) | Running time (500 iterations) | Running time (1000 iterations) | Running time (2000 iterations) |
|---|---|---|---|---|
| Without pre-computation | 77.71 mins (±6.03 mins) | 270.88 mins (±5.11 mins) | 541.76 mins (±10.21 mins) | 1083.52 mins (±20.42 mins) |
| INSTINCT (with pre-computation) | 29.05 mins (±0.54 mins) | 80.13 mins (±0.17 mins) | 156.37 mins (±0.24 mins) | 308.86 mins (±0.38 mins) |
| Speedup | 2.68 times | 3.38 times | 3.46 times | 3.50 times |

In Table 10, "Without pre-computation" is the baseline whose only difference with our `INSTINCT` algorithm is that this baseline uses forward pass to compute every hidden representation in each iteration, and "INSTINCT (with pre-computation)" is our algorithm. The first column in Table 10 (165 iterations, same as the experiments in our paper) shows that pre-computation drastically reduces the running time of our `INSTINCT` algorithm from 78 mins to 29 mins. In addition, Table 10 also shows that in tasks that require more iterations, the pre-computation brings even more significant speed-ups. This is because as the number of iterations increases, the cost of pre-computation is further amortized across more iterations, which makes the computational savings offered by pre-computation more pronounced.

We perform additional experiments to empirically compare the running time of InstructZero and our `INSTINCT`. The results (Table 11) show that our `INSTINCT` has comparable computational costs with InstructZero.

Table 11: The running time (± standard error) of InstructZero and our `INSTINCT`. The results are averaged over the 20 instruction induction tasks in Table 1.

| Algorithm | Total running time | Time for querying ChatGPT |
|---|---|---|
| InstructZero | 18.00 mins (±0.41 mins) | 11.97 mins (±0.35 mins) |
| INSTINCT (including pre-computation) | 29.05 mins (±0.54 mins) | 11.99 mins (±0.32 mins) |

Our `INSTINCT` is slightly slower than InstructZero due to the use of the neural network and the pre-computation of the transformer representations. However, with this slight increase in the computational cost, our `INSTINCT` has achieved significantly better performances than InstructZero as demonstrated in Table 1, Table 3, Table 2, and Table 4. Moreover, as shown in Table 10, the method of pre-computation is crucial for our `INSTINCT` to achieve a comparable computational cost with InstructZero, and the benefit of the pre-computation becomes more significant as a larger query budget is adopted. This further demonstrates the importance of the contribution of our method of pre-computation.

# G    ADDTIONAL COMPARISON WITH THE EVOLUTIONARY ALGORITHM

Recent concurrent works (e.g., EvoPrompt (Guo et al., 2023) and PROMPTBREEDER (Fernando et al., 2023)) use the evolutionary algorithm to find the best instructions. To further demonstrate our superior performance over these evolutionary algorithm-based methods, we perform additional experiments to compare with the concurrent work: EvoPrompt (released on arXiv only on Sep 15, 2023). The results (Table 12) show that our `INSTINCT` significantly outperforms the concurrent work of EvoPrompt, even though EvoPrompt requires more queries to ChatGPT. To elaborate, EvoPrompt uses more queries to ChatGPT than our `INSTINCT`. This is because, in every iteration, Evoprompt needs to query ChatGPT to generate a new instruction and query ChatGPT again to obtain its score, whereas our `INSTINCT` only needs the latter query to ChatGPT (to obtain the score). Despite of getting less feedbacks from the black-box LLM (i.e., ChatGPT), our `INSTINCT` still performs better than EvoPrompt. This is because our `INSTINCT` is based on neural bandits which

Table 12: Average test accuracy (standard error) achieved by the best instruction discovered by different algorithms (including the new evolutionary algorithm-based baseline: EvoPrompt) for different tasks (3 independent trials with different random seeds). The results including all tasks are given in Table 1.

| Task | APE | InstructZero | EvoPrompt | INSTINCT (ours) |
|---|---|---|---|---|
| antonyms | 0.6367(0.1416) | 0.8267(0.0072) | 0.7967(0.0152) | **0.8467(0.0027)** |
| auto_categorization | 0.2500(0.0094) | 0.2567(0.0119) | **0.2600(0.0309)** | 0.2500(0.0330) |
| auto_debugging | 0.2917(0.0340) | **0.3750(0.0000)** | **0.3750(0.0000)** | 0.2917(0.0340) |
| cause_and_effect | 0.5733(0.0891) | 0.8133(0.0109) | **0.8267(0.0475)** | 0.5867(0.0871) |
| common_concept | 0.0691(0.0207) | 0.0864(0.0398) | 0.1211(0.0000) | **0.2129(0.0019)** |
| diff | 0.6733(0.2667) | 0.6933(0.2224) | **1.0000(0.0000)** | **1.0000(0.0000)** |
| informal_to_formal | 0.5736(0.0026) | 0.5310(0.0024) | **0.6182(0.0047)** | 0.5534(0.0000) |
| letters_list | **1.0000(0.0000)** | 0.5900(0.1674) | **1.0000(0.0000)** | **1.0000(0.0000)** |
| negation | 0.7533(0.0109) | 0.7767(0.0136) | 0.7867(0.0027) | **0.8167(0.0027)** |
| object_counting | **0.3633(0.0191)** | 0.3600(0.0929) | 0.1167(0.0626) | 0.3400(0.0698) |
| odd_one_out | 0.6333(0.0144) | 0.6133(0.0871) | 0.6533(0.0109) | **0.7000(0.0163)** |
| orthography_starts_with | 0.4567(0.1477) | 0.5067(0.0871) | 0.6000(0.0205) | **0.6667(0.0272)** |
| rhymes | 0.1567(0.0640) | **1.0000(0.0000)** | 0.6133(0.0178) | **1.0000(0.0000)** |
| second_word_letter | **0.7467(0.2028)** | 0.4333(0.1872) | 0.4133(0.2397) | 0.1000(0.0411) |
| sentence_similarity | 0.0000(0.0000) | 0.0000(0.0000) | **0.2833(0.0357)** | 0.1400(0.0047) |
| sum | 0.6733(0.2667) | **1.0000(0.0000)** | **1.0000(0.0000)** | **1.0000(0.0000)** |
| synonyms | **0.3600(0.0759)** | 0.2767(0.0925) | 0.1367(0.0054) | 0.3067(0.0491) |
| taxonomy_animal | 0.3467(0.2341) | 0.7167(0.0838) | 0.7167(0.1308) | **0.8567(0.0599)** |
| word_sorting | 0.3300(0.0374) | 0.3100(0.1143) | **0.5167(0.0435)** | 0.5133(0.0027) |
| word_unscrambling | 0.4400(0.1389) | 0.5500(0.0170) | 0.6033(0.0072) | **0.6333(0.0072)** |
| # best-performing tasks | 4 | 3 | 9 | 11 |
| average rank | 3.05 | 2.6 | 1.95 | 1.8 |

is a global optimization algorithm that is able to utilize the historical data to efficiently balance exploration vs. exploitation. In contrast, EvoPrompt uses evolutionary algorithm which is a local optimization algorithm that cannot utilize historical data to efficiently explore the global search space of instructions. Moreover, some other concurrent works on black-box methods for instruction optimization (e.g., PROMPTBREEDER (Fernando et al., 2023)) are based on similar core ideas as EvoPrompt, i.e., they also use a powerful LLM (e.g., ChatGPT) to propose new candidate instructions via rephrasing and use evolutionary algorithm for instruction optimization. Therefore, we expect our performance advantage over EvoPrompt (which is the most relevant work to our setting and the most recent of these works to the best of our knowledge) to also hold for the other black-box methods.

## H   PERFORMANCE PROFILE CURVE FOR INSTRUCTION INDUCTION TASKS

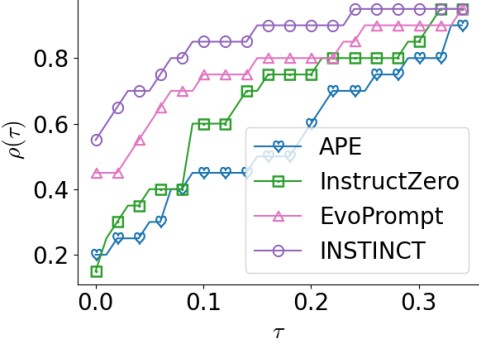

Figure 16: The performance profile curve for the 20 instruction induction tasks in Table 1.

The performance profile curve (Dolan & Moré, 2002) shows how frequently the performance of different approaches is within a certain distance of the best performance. To draw the performance profile curve for a method, for each task $i$, we check whether the performance of this method in task $i$ is within $\tau$ distance to the best performance (among different methods) in task $i$, and hence define an indicator function $\mathbb{I}()$. Next, we average this indicator function across all $n_p$ tasks, which yields a

value $\rho(\tau)$ (equation 5). Finally, the performance profile curve for this method is obtained by varying the value of $\tau$ and calculating the corresponding $\rho(\tau)$.

$$\rho(\tau) = \frac{\sum_{i=1}^{n_p} \mathbb{I}\big((\text{Best performance of task i} - \text{Performance of the approach on task i}) \leq \tau\big)}{n_p} \quad (5)$$

Fig. 16 shows the performance profile for APE, InstructZero, and `INSTINCT`. `INSTINCT` consistently performs the best since it always has the highest $\rho(\tau)$ under different $\tau$. The performance profile curves also make it easier to visualize the performance superiority of our `INSTINCT` algorithm.

## H.1 Peformance of different combinations of white-box LLMs and black-box LLMs

We conduct further experiments to show that our approach is able to generalize to different combinations of black-box LLM and white-box LLM. Specifically, we further consider the black-box LLM: PaLM2 (Anil et al., 2023)[3], GPT4 and white-box LLM: WizardLM (Xu et al., 2023). We compare six combinations: GPT3.5+Vicuna, PaLM2+Vicuna, GPT3.5+WizardLM, PaLM2+WizardLM, GPT4+Vicuna and GPT4+WizardLM. The number of soft tokens is set to 3 and we vary the intrinsic dimension among $[10, 50, 100]$ to obtain the best performance. Table 13 shows the results for different combinations.

Table 13: Test accuracy for different combinations of black-box LLMs + white-box LLMs using `INSTINCT`.

| Black-box LLM | GPT3.5 | PaLM2 | GPT3.5 | PaLM2 | GPT4 | GPT4 |
|---|---|---|---|---|---|---|
| White-box LLM | Vicuna | Vicuna | WizardLM | WizardLM | Vicuna | WizardLM |
| antonyms | 0.8200 | 0.8200 | **0.8400** | 0.8300 | 0.8300 | 0.8000 |
| auto_categorization | 0.1600 | 0.2200 | 0.2600 | 0.2100 | 0.1500 | **0.3400** |
| auto_debugging | **0.3750** | **0.3750** | **0.3750** | **0.3750** | 0.2500 | 0.2500 |
| cause_and_effect | 0.8000 | **1.0000** | 0.5600 | **1.0000** | 0.9600 | 0.8800 |
| common_concept | **0.1959** | 0.0312 | 0.1103 | 0.1688 | 0.1107 | 0.1562 |
| diff | 0.6700 | 0.9700 | **1.0000** | **1.0000** | 0.9800 | **1.0000** |
| informal_to_formal | 0.5534 | 0.4950 | **0.6071** | 0.5372 | 0.5594 | 0.4596 |
| letters_list | **1.0000** | 0.9900 | **1.0000** | 0.9300 | **1.0000** | **1.0000** |
| negation | 0.7600 | 0.7900 | 0.8000 | **0.8400** | 0.8100 | 0.7600 |
| object_counting | 0.3300 | **0.6500** | 0.2800 | 0.6000 | 0.5700 | **0.6500** |
| odd_one_out | 0.7400 | 0.6200 | 0.7400 | 0.6200 | **0.7800** | **0.7800** |
| orthography_starts_with | 0.4700 | 0.4800 | 0.7100 | 0.5200 | 0.6700 | **0.7200** |
| rhymes | **1.0000** | 0.9900 | 0.6100 | 0.8100 | **1.0000** | 0.9900 |
| second_word_letter | 0.1400 | 0.2000 | 0.3500 | 0.2300 | **0.8800** | 0.7400 |
| sentence_similarity | 0.0000 | 0.0000 | 0.0000 | **0.1900** | 0.0000 | 0.1100 |
| sum | **1.0000** | **1.0000** | **1.0000** | **1.0000** | **1.0000** | **1.0000** |
| synonyms | 0.3400 | 0.3600 | 0.1700 | 0.2000 | **0.4700** | 0.1400 |
| taxonomy_animal | 0.8900 | 0.8400 | 0.9300 | 0.8900 | 0.9500 | **1.0000** |
| word_sorting | 0.2700 | 0.0700 | 0.5400 | 0.1900 | 0.6000 | **0.7100** |
| word_unscrambling | 0.6500 | 0.1200 | 0.6100 | 0.1900 | **0.6900** | 0.6700 |
| Average ranking | 3.5 | 3.85 | 3.0 | 3.15 | 2.45 | 2.65 |

In addition to showing that our `INSTINCT` algorithm performs consistently well across different combinations, the results in Table 13 also provide some additional interesting insights. Firstly, GPT4+Vicuna achieves the best performance followed by GPT4+WizardLM, which is reasonable because GPT4 is the state-of-the-art LLM. Secondly, using WizardLM as the white-box LLM in general leads to better performances than using Vicuna (i.e., GPT3.5+WizardLM is better than GPT3.5+Vicuna, and PaLM2+WizardLM is better than PaLM2+Vicuna), even though WizardLM and Vicuna have the same number of parameters (i.e., 13B). This can also be justified because it is corroborated by some LLM leaderboards (Li et al., 2023; Huggingface, 2023), in which WizardLM ranks higher than Vicuna. Thirdly, using GPT3.5 as the black-box LLM generally leads

---

[3]We use text-bison-001 which is based on PaLM2

Table 14: Test accuracy for the best instruction generated by random selection of 40 soft prompts and the instruction generated by `INSTINCT`.

| Task | Initialization only | `INSTINCT` (ours) |
|------|:---:|:---:|
| antonyms | **0.8500** | 0.8467 |
| auto_categorization | 0.0100 | **0.2500** |
| auto_debugging | 0.2500 | **0.2917** |
| cause_and_effect | 0.5200 | **0.5867** |
| common_concept | 0.0045 | **0.2129** |
| diff | **1.0000** | **1.0000** |
| informal_to_formal | 0.5394 | **0.5534** |
| letters_list | **1.0000** | **1.0000** |
| negation | 0.7600 | **0.8167** |
| object_counting | 0.2200 | **0.3400** |
| odd_one_out | 0.6400 | **0.7000** |
| orthography_starts_with | 0.4400 | **0.6667** |
| rhymes | 0.4900 | **1.0000** |
| second_word_letter | **0.1300** | 0.1000 |
| sentence_similarity | 0.0000 | **0.1400** |
| sum | **1.0000** | **1.0000** |
| synonyms | **0.4200** | 0.3067 |
| taxonomy_animal | **0.9300** | 0.8567 |
| word_sorting | 0.0400 | **0.5133** |
| word_unscrambling | 0.5800 | **0.6333** |
| # best-performing tasks | 7 | 16 |

to better performances than using PaLM2 (which can be seen by comparing GPT3.5+Vicuna with PaLM2+Vicuna, and comparing GPT3.5+WizardLM with PaLM2+WizardLM), which is consistent with the LLM leaderboard in LMSYS (2023).

## I    PERFORMANCE GAIN WITH RESPECT TO RANDOM INITIALIZATION

We compare the instruction generated by `INSTINCT` with the best instruction generated by random selection of 40 and 165 soft prompts to show the improvement of our algorithm over random selection. The results are in Table 14 and Table 15. The results show that our `INSTINCT` consistently outperforms these two pure exploration baselines. This is because our `INSTINCT` effectively balances exploration vs. exploitation during the optimization process.

## J    COMPARISON WITH OTHER MODERN BO STRATEGIES

We conduct additional experiments on using other modern BO strategies: BORE (Tiao et al., 2021) and TuRBO (Eriksson et al., 2019) to modify the BO strategy used in InstructZero and show that our `INSTINCT` algorithm performs significantly better than BORE and TuRBO. Specifically, we further compare BORE and TuRBO which are two of the state-of-the-art modern BO strategies. Note that TPE (Bergstra et al., 2011) is compared in the BORE paper and BORE outperforms TPE by a large margin. Therefore, we will not compare TPE here.

From Table 16, we can see that our `INSTINCT` algorithm performs the best and is significantly better than other BO strategies, i.e., InstructZero, InstructZero (BORE), InstructZero (TuRBO). Specifically, InstructZero (BORE) performs better than the original InstructZero, this is because BORE enjoys an improved expressiveness by casting the computation of EI as a binary classification problem

Table 15: Test accuracy for the best instruction generated by random selection of 165 soft prompts and the instruction generated by INSTINCT.

| Task | 165 random | INSTINCT (ours) |
|---|---|---|
| antonyms | 0.8067 | **0.8467** |
| auto_categorization | 0.2033 | **0.2500** |
| auto_debugging | **0.2917** | **0.2917** |
| cause_and_effect | 0.5467 | **0.5867** |
| common_concept | 0.0196 | **0.2129** |
| diff | 0.2533 | **1.0000** |
| informal_to_formal | 0.4200 | **0.5534** |
| letters_list | **1.0000** | **1.0000** |
| negation | 0.6733 | **0.8167** |
| object_counting | **0.3533** | 0.3400 |
| odd_one_out | 0.6733 | **0.7000** |
| orthography_starts_with | 0.5233 | **0.6667** |
| rhymes | 0.6000 | **1.0000** |
| second_word_letter | **0.1167** | 0.1000 |
| sentence_similarity | 0.0133 | **0.1400** |
| sum | 0.9433 | **1.0000** |
| synonyms | 0.2300 | **0.3067** |
| taxonomy_animal | **0.9367** | 0.8567 |
| word_sorting | 0.2400 | **0.5133** |
| word_unscrambling | 0.5867 | **0.6333** |
| # best-performing tasks | 5 | 17 |

Table 16: Average test accuracy (standard error) achieved by the best instruction discovered by different algorithms (including the new modern BO strategies: BORE and TuRBO) for different tasks (3 independent trials with different random seeds). The results including all tasks are given in Table 1.

| Task | APE | InstructZero | InstructZero (TuRBO) | InstructZero (BORE) | EvoPrompt | INSTINCT (ours) |
|---|---|---|---|---|---|---|
| antonyms | 0.6367(0.1416) | 0.8267(0.0072) | 0.8400(0.0100) | **0.8467(0.0067)** | 0.7967(0.0152) | 0.8467(0.0027) |
| auto_categorization | 0.2500(0.0094) | 0.2567(0.0119) | 0.1333(0.1135) | **0.2633(0.0033)** | 0.2600(0.0309) | 0.2500(0.0330) |
| auto_debugging | 0.2917(0.0340) | **0.3750(0.0000)** | 0.2917(0.0417) | 0.2917(0.0417) | **0.3750(0.0000)** | 0.2917(0.0340) |
| cause_and_effect | 0.5733(0.0891) | 0.8133(0.0109) | 0.6400(0.1442) | 0.6400(0.1007) | **0.8267(0.0475)** | 0.5867(0.0871) |
| common_concept | 0.0691(0.0207) | 0.0864(0.0398) | 0.1122(0.0371) | 0.1280(0.0576) | 0.1211(0.0000) | **0.2129(0.0019)** |
| diff | 0.6733(0.2667) | 0.6933(0.2224) | 0.5567(0.2774) | 0.6500(0.2829) | **1.0000(0.0000)** | **1.0000(0.0000)** |
| informal_to_formal | 0.5736(0.0026) | 0.5310(0.0024) | 0.3341(0.1454) | 0.4941(0.0275) | **0.6182(0.0047)** | 0.5534(0.0000) |
| letters_list | **1.0000(0.0000)** | 0.5900(0.1674) | 0.6200(0.2610) | **1.0000(0.0000)** | **1.0000(0.0000)** | **1.0000(0.0000)** |
| negation | 0.7533(0.0109) | 0.7767(0.0136) | 0.7867(0.0384) | 0.7800(0.0603) | 0.7867(0.0027) | **0.8167(0.0027)** |
| object_counting | 0.3633(0.0191) | 0.3600(0.0929) | 0.3667(0.0484) | **0.4167(0.0233)** | 0.1167(0.0626) | 0.3400(0.0698) |
| odd_one_out | 0.6333(0.0144) | 0.6133(0.0871) | 0.5333(0.1775) | 0.5000(0.1747) | 0.6533(0.0109) | **0.7000(0.0163)** |
| orthography_starts_with | 0.4567(0.1477) | 0.5067(0.0871) | **0.6933(0.0219)** | 0.5067(0.1067) | 0.6000(0.0205) | 0.6667(0.0272) |
| rhymes | 0.1567(0.0640) | **1.0000(0.0000)** | **1.0000(0.0000)** | 0.5800(0.2139) | 0.6133(0.0178) | **1.0000(0.0000)** |
| second_word_letter | **0.7467(0.2028)** | 0.4333(0.1872) | 0.4100(0.2902) | 0.2267(0.0318) | 0.4133(0.2397) | 0.1000(0.0411) |
| sentence_similarity | 0.0000(0.0000) | 0.0000(0.0000) | 0.0400(0.0400) | 0.0167(0.0167) | **0.2833(0.0357)** | 0.1400(0.0047) |
| sum | 0.6733(0.2667) | **1.0000(0.0000)** | 0.9867(0.0133) | **1.0000(0.0000)** | **1.0000(0.0000)** | **1.0000(0.0000)** |
| synonyms | 0.3600(0.0759) | 0.2767(0.0925) | 0.3433(0.0536) | **0.3833(0.0067)** | 0.1367(0.0054) | 0.3067(0.0491) |
| taxonomy_animal | 0.3467(0.2341) | 0.7167(0.0838) | 0.7300(0.1242) | **0.8600(0.0693)** | 0.7167(0.1308) | 0.8567(0.0599) |
| word_sorting | 0.3300(0.0374) | 0.3100(0.1143) | 0.2200(0.1266) | 0.2567(0.1087) | **0.5167(0.0435)** | 0.5133(0.0027) |
| word_unscrambling | 0.4400(0.1389) | 0.5500(0.0170) | 0.4533(0.0581) | 0.5567(0.0088) | 0.6033(0.0072) | **0.6333(0.0072)** |
| average rank | 4.25 | 3.55 | 3.85 | 3.05 | 2.55 | 2.35 |

and further uses an MLP as the classifier, and hence BORE is a better BO strategy in instruction optimization. Since the BORE paper already shows that BORE performs significantly better than TPE, we believe that our INSTINCT will also perform significantly better than TPE. InstructZero (TuRBO) performs slightly worse than InstructZero, this is because the InstructZero adopts the instruction-coupled Kernel that uses the instruction scores to improve the kernel function which cannot be easily adopted by TuRBO.

