# OpenReview forum: "Use Your INSTINCT: INSTruction optimization usIng Neural bandits Coupled with Transformers"
_ICLR.cc/2024/Conference — Submitted to ICLR 2024_

### Official Review · Reviewer_fbJ6 · 2023-10-31

**Soundness:** 3 good
**Presentation:** 3 good
**Contribution:** 2 fair
**Rating:** 6
**Confidence:** 4

**Summary:**

Large language models (LLMs) have shown remarkable performance on many tasks, mainly due to their strong instruction-following capabilities. However, their performance depends heavily on the instructions/prompts given to them. Manually designing good instructions is human-intensive and costly. Thus, developing methods to automatically optimize instructions for LLMs is important. Recent works propose gradient-free methods for prompt tuning. Concretely, Chen et al. propose InstructZero which uses Bayesian optimization with Gaussian processes to find optimal soft-prompts that can be used to generate optimal prompts.
Soft prompts are vectors fed to an open-source LLM, which can generate a human-readable and task-relevant instruction given a few exemplars of the target task. The instruction is then submitted to the black-box LLM for evaluation on the target task, whose performance is used to guide the optimization of the soft prompt toward generating better instructions.
However, GPs struggle to model complex high-dimensional functions like the LLM score function.This work proposes INSTINCT, which replaces the Gaussian process with a natural bandit (Neural Upper Confidence Bound - NeuralUCB (Zhou et al., 2020)) based on a pre-trained LLM to improve modeling capability. This allows efficient exploration vs exploitation for optimizing black-box LLM instructions. Experiments show INSTINCT consistently outperforms existing methods on instruction optimization tasks for the black-box model ChatGPT.

**Strengths:**

1. Authors tackle an important problem in leveraging large language models - automatically optimizing instructions/prompts to get better performance from LLMs. Manual prompt design is costly. They clearly identify limitations of prior work on prompt tuning that relies on GPs as such models struggle to find the optimum in high dimensional spaces such as prompt tuning.
2. Authors improve the previous results of InstructZero (INSTINCT) and replace the Gaussian process with a neural bandit model based on pre-trained LLM embeddings, improving modeling capability for the LLM score function. Neural bandits based on pre-trained embeddings are sample efficient due to the power of pre-trained models used for embeddings.
3. The paper is well written and easy to follow, most notably authors clearly explain the prior work and outline their contributions. They highlight the limitations of prior work, the proposed method, and present convincing experimental results.
4. Authors  evaluate their method on instruction optimization tasks for ChatGPT, compare it to prior state-of-the-art methods, and show consistent improvements. They run ablation studies to explore the strength of using pre-trained LLM-based representations of the prompts which they show offers good similarity measure for prompts.

**Weaknesses:**

1. Authors claim the largest problem of GP based BO is how GPs cannot deal with high dimensional optimization problems. It is unclear how this affects automated tuning of Prompts. I would love to see how:  increasing the dimensionality D of the soft prompt affects the accuracy of the downstream accuracy of the black-box LLM with the best prompt chosen based on D. This should be repeated for both normal GP based BO and the NN-bandits based BO to see how the curves change.

2. Authors improve on InstructZero by exploring different surrogate models in the BO scheme introduced by the original authors. Neural Network based bandits are shown to be an efficient mechanism but it is unclear whether a different surrogate function would not offer larger improvements. Similarly, more discussion about acquisition functions would be important. Overall, Authors choose one BO strategy, different than InstructZero and it is unclear whether it is better than any other, modern BO strategy, perhaps BORE [https://proceedings.mlr.press/v139/tiao21a.html] TPE [https://proceedings.neurips.cc/paper_files/paper/2011/file/86e8f7ab32cfd12577bc2619bc635690-Paper.pdf] or evolutionary algorithms such as REA [https://arxiv.org/abs/1802.01548].

3.The authors explore the BO approach for prompt tuning but neglect the recent line of work of evolutionary methods for generating better prompts, for example PROMPTBREEDER [https://arxiv.org/pdf/2309.16797.pdf]. I believe it would be helpful to see how this strategy fares against others. I understand the paper I have references only came out recently but nevertheless more extensive evaluation is needed.

4.The empirical evaluation is limited to optimizing instructions for a single black-box LLM (ChatGPT) and a single white-box LLM (Vicuna). Testing on more LLMs would strengthen the results.

**Questions:**

What is the main motivation behind using NN bandits for this problem instead of other BO methods?

---

> ### Author Response · Authors · 2023-11-18
> **Response to Reviewer fbJ6 - Part 1/4**
>
> We would like to thank the reviewer for taking the time to review our paper and providing such a detailed and insightful review. We particularly appreciate the reviewer for acknowledging that we tackle an important and clear problem, that our paper is well-written and easy to follow, and that our experimental results are convincing and show consistent improvements.
>
> ---
>
> > "I would love to see how: increasing the dimensionality D of the soft prompt affects the accuracy of the downstream accuracy of the black-box LLM with the best prompt chosen based on D. This should be repeated for both normal GP based BO and the NN-bandits based BO to see how the curves change."
>
> As suggested, we have added experiments to study the impact of the dimensionality of the soft prompt on both InstructZero (GP-based BO) and our INSTINCT (neural bandits-based BO). The results (Response Table 1) show that **GP-based BO (InstructZero) suffers from monotonic performance degradation when the soft prompt dimensionality increases**. In contrast, **neural bandits-based BO (our INSTINCT) maintains comparable performances across different soft prompt dimensionalities**.
>
> *Response Table 1: The fraction of tasks that one approach achieves the best performance among 30 instruction induction tasks under different soft prompt dimensionalities. Higher is better.*
>
> |  Soft prompt dimensions    |   15360 |    25600 |    51200 |
> |:-------------|--------:|---------:|---------:|
> | InstructZero |     0.7 | 0.633333 | 0.566667 |
> | INSTINCT     |     0.6 | 0.7      | 0.633333 |
>
> We additionally present the test accuracy on some specific tasks in Response Tables 2, 3, 4, 5, and 6 below. The results show that the performance of InstructZero declines significantly when the soft prompt dimensionality is 51200, whereas **the performance of our INSTINCT remains consistent**.
>
> *Response Table 2: Test accuracy on informal_to_formal under different soft prompt dimensionalities.*
>
> |   Soft prompt dimensions  |    15360 |    25600 |    51200 |
> |:-------------|---------:|---------:|---------:|
> | InstructZero | 0.536239 | 0.536956 | 0.406201 |
> | INSTINCT     | 0.553387 | 0.525643 | 0.534401 |
>
> *Response Table 3: Test accuracy on taxonomy_animal under different soft prompt dimensionalities.*
>
> |   Soft prompt dimensions  |   15360 |   25600 |   51200 |
> |:-------------|--------:|--------:|--------:|
> | InstructZero |    0.76 |    0.91 |    0.55 |
> | INSTINCT     |    0.89 |    0.93 |    0.93 |
>
> *Response Table 4: Test accuracy on word_sorting under different soft prompt dimensionalities.*
>
> |   Soft prompt dimensions  |   15360 |   25600 |   51200 |
> |:-------------|--------:|--------:|--------:|
> | InstructZero |    0.33 |    0.45 |    0.1  |
> | INSTINCT     |    0.27 |    0.51 |    0.45 |
>
> *Response Table 5: Test accuracy on word_unscrambling under different soft prompt dimensionalities.*
>
> |   Soft prompt dimensions   |   15360 |   25600 |   51200 |
> |:-------------|--------:|--------:|--------:|
> | InstructZero |    0.56 |    0.54 |    0.52 |
> | INSTINCT     |    0.65 |    0.58 |    0.59 |
>
> *Response Table 6: Test accuracy on negation under different soft prompt dimensionalities.*
>
> |   Soft prompt dimensions |   15360 |   25600 |   51200 |
> |:-------------|--------:|--------:|--------:|
> | InstructZero |    0.81 |    0.81 |    0.8  |
> | INSTINCT     |    0.76 |    0.82 |    0.82 |
>
> Thank you for the insightful suggestion. We have added these results to the revised version of our paper.
>
> ---
> ↓↓↓ Continue below ↓↓↓

---

> ### Author Response · Authors · 2023-11-18
> **Response to Reviewer fbJ6 - Part 2/4**
>
> >  Neural Network based bandits are shown to be an efficient mechanism but it is unclear whether a different surrogate function would not offer larger improvements. Similarly, more discussion about acquisition functions would be important. Overall, Authors choose one BO strategy, different than InstructZero and it is unclear whether it is better than any other, modern BO strategy, perhaps BORE TPE or evolutionary algorithms such as REA.
>
> > What is the main motivation behind using NN bandits for this problem instead of other BO methods?
>
> The reason why we have used neural networks as the **surrogate function** and adopted the neural bandits-based **BO strategy** is because they (1) have been shown to be effective in optimizing sophisticated high-dimensional objective functions (Dai et al., 2022), and (2) allow us to **leverage the strong expressive power of pre-trained transformers for instruction optimization**.
>
> Firstly, the neural bandits-based BO replaces the GP surrogate function with a neural network, which has been repeatedly shown to be effective in modeling complicated and high-dimensional objective functions.
> Secondly, the use of neural bandits allows us to stack an MLP on top of the hidden representation from a pre-trained transformer to perform the task of score prediction, which is used during instruction optimization. This paradigm for prediction tasks (hidden representation from pre-trained transformer + MLP) has been repeatedly proven successful (Radford et al., 2018; 2019), and **our proposed INSTINCT algorithm allows us to adopt this successful paradigm for prediction tasks for instruction optimization**.
> These are the unique and significant advantages offered by the neural network surrogate function and the neural bandits-based BO strategy we have adopted, compared with other modern BO strategies. We have also revised the paper to discuss these advantages of our method over the other BO strategies you have referenced (revised Appendix B.3).
>
> Regarding the **acquisition function**, we have adopted the natural choice of the NeuralUCB acquisition function in our INSTINCT algorithm. It is indeed interesting to explore in future works whether other acquisition functions for neural bandits such as Neural Thompson sampling (NeuralTS) (Zhang et al., 2021) can bring further performance improvements as you have suggested.
>
>
> ---
> ↓↓↓ Continue below ↓↓↓

---

> ### Author Response · Authors · 2023-11-18
> **Response to Reviewer fbJ6 - Part 3/4**
>
> > "The authors explore the BO approach for prompt tuning but neglect the recent line of work of evolutionary methods ... for example PROMPTBREEDER ... how this strategy fares against others. I understand the paper I have references only came out recently"
>
> Thank you for the suggestion and for pointing out the concurrent related work of PROMPTBREEDER. We have added reference to PROMPTBREEDER (Fernando et al., 2023) as a concurrent work, since as you have also mentioned, the paper only came out recently (only released on arixv on Sep 28). The code for PROMPTBREEDER has not been released yet and we have found it infeasible to implement the algorithm within the short rebuttal period. Therefore, we have instead added comparisons with another evolutionary algorithm-based concurrent work: EvoPrompt (Guo et al., 2023), which was released on arXiv only on Sep 15. The results (Response Table 7 below) show that **our INSTINCT significantly outperforms the concurrent work of EvoPrompt, even though EvoPrompt requires more queries to ChatGPT** (more details below).
>
> *Response Table 7: Additional comparison with EvoPrompt.*
>
> | Algorithm | APE | InstructZero | EvoPrompt | INSTINCT |
> |:-------------|------:|-------:|------:|----------------------:|
> |antonyms|0.6367(0.1416)|0.8267(0.0072)|0.7967(0.0152)|**0.8467(0.0027)**|
> |auto_categorization|0.2500(0.0094)|0.2567(0.0119)|**0.2600(0.0309)**|0.2500(0.0330)|
> |auto_debugging|0.2917(0.0340)|**0.3750(0.0000)**|**0.3750(0.0000)**|0.2917(0.0340)|
> |cause_and_effect|0.5733(0.0891)|0.8133(0.0109)|**0.8267(0.0475)**|0.5867(0.0871)|
> |common_concept|0.0691(0.0207)|0.0864(0.0398)|0.1211(0.0000)|**0.2129(0.0019)**|
> |diff|0.6733(0.2667)|0.6933(0.2224)|**1.0000(0.0000)**|**1.0000(0.0000)**|
> |informal_to_formal|0.5736(0.0026)|0.5310(0.0024)|**0.6182(0.0047)**|0.5534(0.0000)|
> |letters_list|**1.0000(0.0000)**|0.5900(0.1674)|**1.0000(0.0000)**|**1.0000(0.0000)**|
> |negation|0.7533(0.0109)|0.7767(0.0136)|0.7867(0.0027)|**0.8167(0.0027)**|
> |object_counting|**0.3633(0.0191)**|0.3600(0.0929)|0.1167(0.0626)|0.3400(0.0698)|
> |odd_one_out|0.6333(0.0144)|0.6133(0.0871)|0.6533(0.0109)|**0.7000(0.0163)**|
> |orthography_starts_with|0.4567(0.1477)|0.5067(0.0871)|0.6000(0.0205)|**0.6667(0.0272)**|
> |rhymes|0.1567(0.0640)|**1.0000(0.0000)**|0.6133(0.0178)|**1.0000(0.0000)**|
> |second_word_letter|**0.7467(0.2028)**|0.4333(0.1872)|0.4133(0.2397)|0.1000(0.0411)|
> |sentence_similarity|0.0000(0.0000)|0.0000(0.0000)|**0.2833(0.0357)**|0.1400(0.0047)|
> |sum|0.6733(0.2667)|**1.0000(0.0000)**|**1.0000(0.0000)**|**1.0000(0.0000)**|
> |synonyms|**0.3600(0.0759)**|0.2767(0.0925)|0.1367(0.0054)|0.3067(0.0491)|
> |taxonomy_animal|0.3467(0.2341)|0.7167(0.0838)|0.7167(0.1308)|**0.8567(0.0599)**|
> |word_sorting|0.3300(0.0374)|0.3100(0.1143)|**0.5167(0.0435)**|0.5133(0.0027)|
> |word_unscrambling|0.4400(0.1389)|0.5500(0.0170)|0.6033(0.0072)|**0.6333(0.0072)**|
> | -- | -- | -- | -- | -- |
> | Average ranking | 3.05 | 2.6 | 1.95 | 1.8|
>
> Response Table 7 shows that our INSTINCT consistently outperforms all baselines, including the concurrent work of EvoPrompt.
> In addition, note that EvoPrompt uses more queries to ChatGPT than our INSTINCT. This is because, in every iteration, Evoprompt needs to query ChatGPT to generate a new instruction and query ChatGPT again to obtain its score, whereas our INSTINCT only needs the latter query to ChatGPT (to obtain the score). Despite getting less feedback from the black-box LLM (i.e., ChatGPT), our INSTINCT still performs better than EvoPrompt. This is because our INSTINCT is based on neural bandits which is a **global optimization algorithm** that is able to utilize the historical data to efficiently balance exploration vs. exploitation. In contrast, EvoPrompt uses the evolutionary algorithm which is a **local optimization algorithm** that cannot utilize historical data to efficiently explore the global search space of instructions.
> Moreover, some other concurrent works on black-box methods for instruction optimization are based on similar core ideas as EvoPrompt, i.e., they also use a powerful LLM (e.g., ChatGPT) to propose new candidate instructions via rephrasing and use evolutionary algorithms for instruction optimization.
> Therefore, our performance advantage over the representative method of EvoPrompt is also likely to hold for the other black-box methods.
>
>
> On the other hand, our INSTINCT is in fact not mutually exclusive with these methods that use LLM to generate instructions, in the sense that **these methods can be incorporated into our INSTINCT algorithm to further improve our performance**. Specifically, as a proof-of-concept experiment, in the last paragraph of Sec. 5 in our paper, we had proposed a method to further improve our INSTINCT via ChatGPT rephrasing (similar to the rephrasing approaches used in APE and EvoPrompt), and Figure 5 shows that this rephrasing method has further improved the performance of our INSTINCT.
>
> ---
> ↓↓↓ Continue below ↓↓↓

---

> ### Author Response · Authors · 2023-11-18
> **Response to Reviewer fbJ6 - Part 4/4**
>
> > "The empirical evaluation is limited to optimizing instructions for a single black-box LLM (ChatGPT) and a single white-box LLM (Vicuna). Testing on more LLMs would strengthen the results."
>
>
> As you suggested, we have added experiments using multiple combinations of different white-box and black-box LLMs (using the instruction induction tasks in our main paper) and presented the results in Response Table 1 below. The results show that **our INSTINCT algorithm remains effective under different combinations of white-box and black-box LLMs**. Specifically, we have considered 2 white-box LLMs: Vicuna and WizardLM [1], and 3 black-box LLMs: GPT3.5, GPT4, and PaLM2 [2]. We have fixed the number of soft tokens to 3 and used the validation set to perform a grid search over the intrinsic dimension within {10, 50, 100} (see the first paragraph of Sec. 4 for more detail).
>
> *Response Table 1: Test accuracy on instruction induction tasks using different combinations of black-box and white-box LLMs.*
>
> |    Black-box LLM          |   GPT3.5 |   PaLM2 |   GPT3.5 |   PaLM2 |   GPT4 |GPT4 |
> |:-------------|------:|-------:|------:|-------:|-------:|----------------------:|
> |    **White-box LLM**          |   **Vicuna** |   **Vicuna** |   **WizardLM** |   **WizardLM** |   **Vicuna** | **WizardLM** |
> |    -------------          |   ------------- |   ------------- |   ------------- |   ------------- |   ------------- | ------------- |
> |antonyms|0.8200|0.8200|**0.8400**|0.8300|0.8300|0.8000|
> |auto_categorization|0.1600|0.2200|0.2600|0.2100|0.1500|**0.3400**|
> |auto_debugging|**0.3750**|**0.3750**|**0.3750**|**0.3750**|0.2500|0.2500|
> |cause_and_effect|0.8000|**1.0000**|0.5600|**1.0000**|0.9600|0.8800|
> |common_concept|**0.1959**|0.0312|0.1103|0.1688|0.1107|0.1562|
> |diff|0.6700|0.9700|**1.0000**|**1.0000**|0.9800|**1.0000**|
> |informal_to_formal|0.5534|0.4950|**0.6071**|0.5372|0.5594|0.4596|
> |letters_list|**1.0000**|0.9900|**1.0000**|0.9300|**1.0000**|**1.0000**|
> |negation|0.7600|0.7900|0.8000|**0.8400**|0.8100|0.7600|
> |object_counting|0.3300|**0.6500**|0.2800|0.6000|0.5700|**0.6500**|
> |odd_one_out|0.7400|0.6200|0.7400|0.6200|**0.7800**|**0.7800**|
> |orthography_starts_with|0.4700|0.4800|0.7100|0.5200|0.6700|**0.7200**|
> |rhymes|**1.0000**|0.9900|0.6100|0.8100|**1.0000**|0.9900|
> |second_word_letter|0.1400|0.2000|0.3500|0.2300|**0.8800**|0.7400|
> |sentence_similarity|0.0000|0.0000|0.0000|**0.1900**|0.0000|0.1100|
> |sum|**1.0000**|**1.0000**|**1.0000**|**1.0000**|**1.0000**|**1.0000**|
> |synonyms|0.3400|0.3600|0.1700|0.2000|**0.4700**|0.1400|
> |taxonomy_animal|0.8900|0.8400|0.9300|0.8900|0.9500|**1.0000**|
> |word_sorting|0.2700|0.0700|0.5400|0.1900|0.6000|**0.7100**|
> |word_unscrambling|0.6500|0.1200|0.6100|0.1900|**0.6900**|0.6700|
> | --- | --- | --- | --- | --- | --- |  --- |
> | **Average ranking** | 3.5 | 3.85| 3.0  | 3.15| 2.45| 2.65 |
>
> In addition to showing that **our INSTINCT algorithm performs consistently well across different combinations**, the results in Response Table 1 also provide some additional interesting insights. Firstly, GPT4+Vicuna achieves the best performance followed by GPT4+WizardLM, which is reasonable because GPT4 is the state-of-the-art LLM. Secondly, using WizardLM as the white-box LLM in general leads to better performances than using Vicuna (i.e., GPT3.5+WizardLM is better than GPT3.5+Vicuna, and PaLM2+WizardLM is better than PaLM2+Vicuna), even though WizardLM and Vicuna have the same number of parameters (i.e., 13B). This can also be justified because it is corroborated by some LLM leaderboards [3, 4], in which WizardLM ranks higher than Vicuna. Thirdly, using GPT3.5 as the black-box LLM generally leads to better performances than using PaLM2 (which can be seen by comparing GPT3.5+Vicuna with PaLM2+Vicuna, and comparing GPT3.5+WizardLM with PaLM2+WizardLM), which is consistent with the LLM leaderboard in [5].
>
> ---
>
> Thank you again for your detailed and insightful feedback. We hope our clarifications and additional results could improve your evaluation of our work.
>
> ---
>
> [1] Xu, C., Sun, Q., Zheng, K., Geng, X., Zhao, P., Feng, J., ... & Jiang, D. (2023). Wizardlm: Empowering large language models to follow complex instructions. arXiv preprint arXiv:2304.12244.
>
> [2] Anil, R., Dai, A. M., Firat, O., Johnson, M., Lepikhin, D., Passos, A., ... & Wu, Y. (2023). PaLM 2 technical report. arXiv preprint arXiv:2305.10403.
>
> [3] Li, X., Zhang, T., Dubois, Y., Taori, R., Gulrajani, I., Guestrin, C., Liang, P. & Hashimoto, T. B. (2023). AlpacaEval: An Automatic Evaluator of Instruction-following Models [Computer software]. GitHub. https://github.com/tatsu-lab/alpaca_eval
>
> [4] HuggingFaceH4 / open_llm_leaderboard. (2023). Spaces. Retrieved November 16, 2023, from https://huggingface.co/spaces/HuggingFaceH4/open_llm_leaderboard
>
> [5] Chatbot Arena Leaderboard. (2023). Retrieved November 16, 2023, from https://lmsys.org/blog/2023-05-25-leaderboard/

---

> ### Author Response · Authors · 2023-11-21
> **We would like to know if you have any further questions that require additional clarifications.**
>
> Dear Reviewer fbJ6,
>
> Thank you again for your time in reviewing our paper and for asking many valuable questions.
>
> Please let us know if our replies have addressed your concerns. We would be happy to address any further questions you might have within the discussion period.
>
> Best wishes,
>
> Authors

---

> > ### Comment · Reviewer_fbJ6 · 2023-11-22
> >
> > Dear Authors,
> >
> > I am thouroughly impressed by the depth of this rebuttal. I believe you have answered most of my concerns and the extra empirical results significantly strengthen the paper. My main question of "Authors choose a BO strategy different than InstructZero and it is unclear whether (the one they chose) is better than any other, modern BO strategies" is not yet answered but I understand that this would be a somewhat different, more extensive paper about benchmarking different BO startegied for InstructZero. Based on the provided findings I am happy to improve my review to 6.
> >
> > Best

---

> ### Author Response · Authors · 2023-11-23
> **Response to Reviewer fbJ6: New results on comparison with other modern BO strategies**
>
> Thank you very much for your appreciation of the depth of our rebuttal. We are happy to know that most of your concerns are addressed in our previous rebuttal. We would like to resolve the remaining one on comparison with other modern BO approaches as follows:
>
> > My main question of "Authors choose a BO strategy different than InstructZero and it is unclear whether (the one they chose) is better than any other, modern BO strategies" is not yet answered
>
> We conduct additional experiments using other modern BO strategies: BORE and TuRBO [1] to modify the BO strategy used in InstructZero and show that **our INSTINCT algorithm performs significantly better than BORE and TuRBO**. Specifically, we further compare BORE and TuRBO which are two of the state-of-the-art modern BO strategies. Note that TPE is compared in BORE's paper and BORE outperforms TPE by a large margin and thus we will not compare TPE here.
>
> Response Table 1: Test accuracy of the best instruction from different algorithms.
>
> | Algorithm | APE | InstructZero | InstructZero (TuRBO) | InstructZero (BORE) | EvoPrompt | INSTINCT |
> |:-------------|------:|-------:|------:|------:|------:|----------------------:|
> |antonyms|0.6367(0.1416)|0.8267(0.0072)|0.8400(0.0100)|0.8467(0.0067)|0.7967(0.0152)|**0.8467(0.0027)**|
> |auto_categorization|0.2500(0.0094)|0.2567(0.0119)|0.1333(0.1135)|**0.2633(0.0033)**|0.2600(0.0309)|0.2500(0.0330)|
> |auto_debugging|0.2917(0.0340)|**0.3750(0.0000)**|0.2917(0.0417)|0.2917(0.0417)|**0.3750(0.0000)**|0.2917(0.0340)|
> |cause_and_effect|0.5733(0.0891)|0.8133(0.0109)|0.6400(0.1442)|0.6400(0.1007)|**0.8267(0.0475)**|0.5867(0.0871)|
> |common_concept|0.0691(0.0207)|0.0864(0.0398)|0.1122(0.0371)|0.1280(0.0576)|0.1211(0.0000)|**0.2129(0.0019)**|
> |diff|0.6733(0.2667)|0.6933(0.2224)|0.5567(0.2774)|0.6500(0.2829)|**1.0000(0.0000)**|**1.0000(0.0000)**|
> |informal_to_formal|0.5736(0.0026)|0.5310(0.0024)|0.3341(0.1454)|0.4941(0.0275)|**0.6182(0.0047)**|0.5534(0.0000)|
> |letters_list|**1.0000(0.0000)**|0.5900(0.1674)|0.6200(0.2610)|**1.0000(0.0000)**|**1.0000(0.0000)**|**1.0000(0.0000)**|
> |negation|0.7533(0.0109)|0.7767(0.0136)|0.7867(0.0384)|0.7800(0.0603)|0.7867(0.0027)|**0.8167(0.0027)**|
> |object_counting|0.3633(0.0191)|0.3600(0.0929)|0.3667(0.0484)|**0.4167(0.0233)**|0.1167(0.0626)|0.3400(0.0698)|
> |odd_one_out|0.6333(0.0144)|0.6133(0.0871)|0.5333(0.1775)|0.5000(0.1747)|0.6533(0.0109)|**0.7000(0.0163)**|
> |orthography_starts_with|0.4567(0.1477)|0.5067(0.0871)|**0.6933(0.0219)**|0.5067(0.1067)|0.6000(0.0205)|0.6667(0.0272)|
> |rhymes|0.1567(0.0640)|**1.0000(0.0000)**|**1.0000(0.0000)**|0.5800(0.2139)|0.6133(0.0178)|**1.0000(0.0000)**|
> |second_word_letter|**0.7467(0.2028)**|0.4333(0.1872)|0.4100(0.2902)|0.2267(0.0318)|0.4133(0.2397)|0.1000(0.0411)|
> |sentence_similarity|0.0000(0.0000)|0.0000(0.0000)|0.0400(0.0400)|0.0167(0.0167)|**0.2833(0.0357)**|0.1400(0.0047)|
> |sum|0.6733(0.2667)|**1.0000(0.0000)**|0.9867(0.0133)|**1.0000(0.0000)**|**1.0000(0.0000)**|**1.0000(0.0000)**|
> |synonyms|0.3600(0.0759)|0.2767(0.0925)|0.3433(0.0536)|**0.3833(0.0067)**|0.1367(0.0054)|0.3067(0.0491)|
> |taxonomy_animal|0.3467(0.2341)|0.7167(0.0838)|0.7300(0.1242)|**0.8600(0.0693)**|0.7167(0.1308)|0.8567(0.0599)|
> |word_sorting|0.3300(0.0374)|0.3100(0.1143)|0.2200(0.1266)|0.2567(0.1087)|**0.5167(0.0435)**|0.5133(0.0027)|
> |word_unscrambling|0.4400(0.1389)|0.5500(0.0170)|0.4533(0.0581)|0.5567(0.0088)|0.6033(0.0072)|**0.6333(0.0072)**|
> | -- | -- | -- | -- | -- | -- | -- |
> | **Average ranking** | 4.25 | 3.55 | 3.85 | 3.05|2.55|2.35|
>
> From Response Table 1, we can see that **our INSTINCT algorithm performs the best and is significantly better than other BO strategies, i.e., InstructZero, InstructZero (BORE), InstructZero (TuRBO)**. Specifically, InstructZero (BORE) performs better than the original InstructZero, this is because BORE enjoys an improved expressiveness by casting the computation of EI as a binary classification problem and further uses an MLP as the classifier, and hence BORE is a better BO strategy in instruction optimization. Since the BORE paper already shows that BORE performs significantly better than TPE, we believe that our INSTINCT will also perform significantly better than TPE. InstructZero (TuRBO) performs slightly worse than InstructZero, this is because the InstructZero adopts the instruction-coupled kernel that uses the instruction scores to improve the kernel function which cannot be easily adopted by TuRBO.
>
> ---
> Thank you for the insightful suggestion. We have added these results to Appendix J of the revised version of our paper. We hope our clarifications and additional results could improve your evaluation of our work.
>
> [1] Eriksson, D., Pearce, M., Gardner, J., Turner, R. D., & Poloczek, M. (2019). Scalable global optimization via local Bayesian optimization. Advances in neural information processing systems, 32.

---

### Official Review · Reviewer_16Xs · 2023-10-31

**Soundness:** 2 fair
**Presentation:** 2 fair
**Contribution:** 2 fair
**Rating:** 3
**Confidence:** 5

**Summary:**

The paper discusses the dependency of Large Language Models (LLMs) on specific instructions for optimal performance, which are usually manually fine-tuned. It mentions the use of a Bayesian optimization (BO) algorithm to automate instruction optimization but highlights its inadequacy in handling complex, high-dimensional objective functions. To overcome this, the authors introduce a neural bandit algorithm replacing the Gaussian process in BO with a Neural Network (NN) surrogate. This new method, named INSTINCT (INStruction optimization usIng Neural bandits Coupled with Transformers), leverages pre-trained transformers to better model the objective function and optimize instructions. Through extensive experimentation with ChatGPT on various tasks, the INSTINCT algorithm demonstrated superior performance compared to existing methods, showcasing its efficacy in enhancing instruction optimization for black-box LLMs.

**Strengths:**

1. Adopt the NeuralUCB algorithm and propose INSTINCT algorithm to improve the instruction optimization.
2. Conduct comprehensive experiments regarding the tasks.

**Weaknesses:**

1. lack of the novelty: since the NeuralUCB is the existing algorithm and InstructZero is the existing pipeline for optimizing the instructions for black-box models, i.e., ChatGPT. It seems that the idea is just a combination of InstructZero and NeuralUCB.
2. lack of the experiment on different combinations of white-box + block-box model, e.g., GPT4 + WizardLM, GPT4 + Vicuna. I would like to see how the different combinations affect the results.
3. Since the white-box model, Vicuna, is a distilled model from GPT-family, I would like to see how the algorithm can optimize the instruction for black-box models from other families. If you want to claim that your algorithm can generalize well, please conduct these experiments.

**Questions:**

1. how many demos do you use for white-box LLM?

---

> ### Author Response · Authors · 2023-11-18
> **Response to Reviewer 16Xs - Part 1/3**
>
> We would like to thank the reviewer for taking the time to review our paper and providing such a detailed and constructive review. We particularly appreciate the reviewer for acknowledging that our experiments are extensive and comprehensive and that our proposed method is efficacious in enhancing instruction optimization for black-box LLMs.
>
> ---
>
> > "Lack of the novelty: ...  It seems that the idea is just a combination of InstructZero and NeuralUCB."
>
> We would like to clarify that our idea **is far more than just a combination of InstructZero and NeuralUCB**.
>
> Firstly, the strong expressive power of pre-trained transformers is a key reason behind the success of LLMs. However, **how do we leverage the strong expressive power of pre-trained transformers for instruction optimization**? This significant research question remained unanswered till our work, which not only **proposed a theoretically grounded solution to achieve this** but also **empirically verified the resulting significant performance improvement through extensive experiments**. Specifically, the use of neural bandits allows us to stack an MLP on top of the hidden representation from a pre-trained transformer to perform the task of score prediction, which is used during instruction optimization. This paradigm for prediction tasks (pre-trained transformer + MLP) has been repeatedly proven successful (Radford et al., 2018; 2019), and **our proposed INSTINCT algorithm allows us to adopt this successful paradigm for prediction tasks for instruction optimization**, which, to the best of our knowledge, **was not possible before our work**.
>
> Secondly, from the perspective of the field of neural bandits, our proposed INSTINCT algorithm has also made important contributions to this field. Specifically, to the best of our knowledge, **we are the first work (1) to couple neural bandits with the hidden representation from pre-trained transformers, and (2) to apply neural bandits to problems with such high-dimensional input spaces** (the input dimensions in our experiments are in the range of 5120-51200).
>
> Lastly, note that naively (1) coupling neural bandits with the hidden representation from pre-trained transformers and (2) applying neural bandits to such high-dimensional problems will lead to excessively high computational costs. In our work, we have overcome this technical challenge and hence **made our algorithm computationally feasible through our proposed method of pre-computation, which is another novel and important contribution of our work** (Sec. 3.2, the paragraph "Pre-Computation to Save Costs").
>
> ---
> ↓↓↓ Continue below ↓↓↓

---

> ### Author Response · Authors · 2023-11-18
> **Response to Reviewer 16Xs - Part 2/3**
>
> > "Lack of the experiment on different combinations of white-box + black-box model"
>
> As you suggested, we have added experiments using multiple combinations of different white-box and black-box LLMs (using the instruction induction tasks in our main paper) and presented the results in Response Table 1 below. The results show that **our INSTINCT algorithm remains effective under different combinations of white-box and black-box LLMs**. Specifically, we have considered 2 white-box LLMs: Vicuna and WizardLM [1], and 3 black-box LLMs: GPT3.5, GPT4, and PaLM2 [2]. We have fixed the number of soft tokens to 3 and used the validation set to perform a grid search over the intrinsic dimension within {10, 50, 100} (see the first paragraph of Sec. 4 for more detail).
>
> *Response Table 1: Test accuracy on instruction induction tasks using different combinations of black-box and white-box LLMs.*
>
> |    Black-box LLM          |   GPT3.5 |   PaLM2 |   GPT3.5 |   PaLM2 |   GPT4 |GPT4 |
> |:-------------|------:|-------:|------:|-------:|-------:|----------------------:|
> |    **White-box LLM**          |   **Vicuna** |   **Vicuna** |   **WizardLM** |   **WizardLM** |   **Vicuna** | **WizardLM** |
> |    -------------          |   ------------- |   ------------- |   ------------- |   ------------- |   ------------- | ------------- |
> |antonyms|0.8200|0.8200|**0.8400**|0.8300|0.8300|0.8000|
> |auto_categorization|0.1600|0.2200|0.2600|0.2100|0.1500|**0.3400**|
> |auto_debugging|**0.3750**|**0.3750**|**0.3750**|**0.3750**|0.2500|0.2500|
> |cause_and_effect|0.8000|**1.0000**|0.5600|**1.0000**|0.9600|0.8800|
> |common_concept|**0.1959**|0.0312|0.1103|0.1688|0.1107|0.1562|
> |diff|0.6700|0.9700|**1.0000**|**1.0000**|0.9800|**1.0000**|
> |informal_to_formal|0.5534|0.4950|**0.6071**|0.5372|0.5594|0.4596|
> |letters_list|**1.0000**|0.9900|**1.0000**|0.9300|**1.0000**|**1.0000**|
> |negation|0.7600|0.7900|0.8000|**0.8400**|0.8100|0.7600|
> |object_counting|0.3300|**0.6500**|0.2800|0.6000|0.5700|**0.6500**|
> |odd_one_out|0.7400|0.6200|0.7400|0.6200|**0.7800**|**0.7800**|
> |orthography_starts_with|0.4700|0.4800|0.7100|0.5200|0.6700|**0.7200**|
> |rhymes|**1.0000**|0.9900|0.6100|0.8100|**1.0000**|0.9900|
> |second_word_letter|0.1400|0.2000|0.3500|0.2300|**0.8800**|0.7400|
> |sentence_similarity|0.0000|0.0000|0.0000|**0.1900**|0.0000|0.1100|
> |sum|**1.0000**|**1.0000**|**1.0000**|**1.0000**|**1.0000**|**1.0000**|
> |synonyms|0.3400|0.3600|0.1700|0.2000|**0.4700**|0.1400|
> |taxonomy_animal|0.8900|0.8400|0.9300|0.8900|0.9500|**1.0000**|
> |word_sorting|0.2700|0.0700|0.5400|0.1900|0.6000|**0.7100**|
> |word_unscrambling|0.6500|0.1200|0.6100|0.1900|**0.6900**|0.6700|
> | --- | --- | --- | --- | --- | --- |  --- |
> | **Average ranking** | 3.5 | 3.85| 3.0  | 3.15| 2.45| 2.65 |
>
> In addition to showing that **our INSTINCT algorithm performs consistently well across different combinations**, the results in Response Table 1 also provide some additional interesting insights. Firstly, GPT4+Vicuna achieves the best performance followed by GPT4+WizardLM, which is reasonable because GPT4 is the state-of-the-art LLM. Secondly, using WizardLM as the white-box LLM in general leads to better performances than using Vicuna (i.e., GPT3.5+WizardLM is better than GPT3.5+Vicuna, and PaLM2+WizardLM is better than PaLM2+Vicuna), even though WizardLM and Vicuna have the same number of parameters (i.e., 13B). This can also be justified because it is corroborated by some LLM leaderboards [3, 4], in which WizardLM ranks higher than Vicuna. Thirdly, using GPT3.5 as the black-box LLM generally leads to better performances than using PaLM2 (which can be seen by comparing GPT3.5+Vicuna with PaLM2+Vicuna, and comparing GPT3.5+WizardLM with PaLM2+WizardLM), which is consistent with the LLM leaderboard in [5].
>
> ---
>
> [1] Xu, C., Sun, Q., Zheng, K., Geng, X., Zhao, P., Feng, J., ... & Jiang, D. (2023). Wizardlm: Empowering large language models to follow complex instructions. arXiv preprint arXiv:2304.12244.
>
> [2] Anil, R., Dai, A. M., Firat, O., Johnson, M., Lepikhin, D., Passos, A., ... & Wu, Y. (2023). PaLM 2 technical report. arXiv preprint arXiv:2305.10403.
>
> [3] Li, X., Zhang, T., Dubois, Y., Taori, R., Gulrajani, I., Guestrin, C., Liang, P. & Hashimoto, T. B. (2023). AlpacaEval: An Automatic Evaluator of Instruction-following Models [Computer software]. GitHub. https://github.com/tatsu-lab/alpaca_eval
>
> [4] HuggingFaceH4 / open_llm_leaderboard. (2023). Spaces. Retrieved November 16, 2023, from https://huggingface.co/spaces/HuggingFaceH4/open_llm_leaderboard
>
> [5] Chatbot Arena Leaderboard. (2023). Retrieved November 16, 2023, from https://lmsys.org/blog/2023-05-25-leaderboard/
>
>
> ---
> ↓↓↓ Continue below ↓↓↓

---

> ### Author Response · Authors · 2023-11-18
> **Response to Reviewer 16Xs - Part 3/3**
>
> >  "I would like to see how the algorithm can optimize the instruction for black-box models from other families"
>
> As we have discussed in our response above, we have added experiments using different combinations of white-box and black-box LLMs (Response Table 1 above), which include combinations where the white-box LLM is not distilled from the black-box LLM: The combinations of PaLM2+Vicuna and PaLM2+WizardLM.
> The results in Response Table 1 show that for these combinations, our INSTINCT algorithm is still able to find effective instructions with high scores. For example, the average ranking of PaLM2+WizardLM is better than GPT3.5+Vicuna and only slightly worse than GPT3.5+WizardLM. Therefore, these new experimental results suggest that our INSTINCT algorithm can generalize well across a wide range of combinations of white-box and black-box LLMs, not necessarily "from the same family".
>
> > "How many demos do you use for white-box LLM?"
>
> For the instruction induction tasks and the text summarization task (Section 4.1), we have used 5 demos for the white-box LLM to generate the instructions, as shown in Figure 6 in our paper (page 14). For the chain-of-thought instruction optimization (Section 4.2), we have used 3 demos to generate new instructions as shown in Figure 9 in our paper (page 17).
>
> ---
>
> Thank you again for your constructive feedback. We hope our additional results and clarifications could improve your opinion of our paper.

---

> ### Author Response · Authors · 2023-11-21
> **We would like to know if you have any further questions that require additional clarifications.**
>
> Dear Reviewer 16Xs,
>
> Thank you again for your time in reviewing our paper and for asking many valuable questions.
>
> Please let us know if our replies have addressed your concerns. We would be happy to address any further questions you might have within the discussion period.
>
>
>
> Best wishes,
>
> Authors

---

> ### Author Response · Authors · 2023-11-23
> **Last day rebuttal reminder: Thank you for reviewing our paper, we would like to know whether our responses have addressed your concerns**
>
> Dear Reviewer 16Xs,
>
> Today is the last day of rebuttal, and we are eager to know whether our responses have addressed your concerns. To summarize:
>
> 1. We compare all the combinations of white-box and black-box LLMs you have proposed and also other combinations of different LLM families to show that our INSTINCT remains effective.
> 2. We explain how our INSTINCT is far more than just a combination of InstructZero and NeuralUCB.
>
>
> We are happy to address any further questions you might have before the rebuttal ends.
>
> Best wishes,
>
> Authors

---

### Official Review · Reviewer_zDza · 2023-10-31

**Soundness:** 3 good
**Presentation:** 3 good
**Contribution:** 3 good
**Rating:** 8
**Confidence:** 3

**Summary:**

This paper introduces a new instruction tuning technique for LLMs that builds on InstructZero, which is recent work using Bayesian optimization to tune instructions. The technique, called INSTINCT, replaces the Gaussian process objective function surrogate in Bayesian optimization-based instruction tuning with a neural network surrogate (NeuralUCB) -- thus increasing the expressive power that it can support. Finally, the hidden representations in the neural surrogate are combined with pretrained LLM hidden representations. The results show improved zero-shot and chain of thought performance across a variety of tasks.

**Strengths:**

- INSTINCT solves a clear problem -- Bayesian Optimization-based techniques for prompting that involve the use of a Gaussian process for modeling the objective might not be the right tool for high-dimensional or complex objectives (which are both true in the case of prompting). Replacing the Gaussian process model with NeuralUCB just makes sense.
- The authors include a thorough evaluation, and show in Tables 1 and 4 that INSTINCT improves the average ranking over the baselines across a wide variety of tasks.
- While combining NeuralUCB with InstructZero makes sense, it poses a nontrivial challenge of computational inefficiency that the authors address via precomputation. I view this is a potentially solid contribution that can be further strengthened via an empirical evaluation of the computational costs involved.

**Weaknesses:**

- While it seems great that INSTINCT can be sped up via pre-computation, can the authors provide an empirical comparison between the computation costs involved with running INSTINCT compared to its baselines? Ideally for the sake of a fair evaluation, this should also include pre-computation.
- Not a deal breaker, but could more baselines be included? It seems that INSTINCT is only compared to two baselines: APE and InstructZero. I do see that this is addressed in Section 6 -- but can any of the black-box methods be adapted to your setting in some simple way?

**Questions:**

- It is interesting that the method can be sped up via pre-computation. Can the authors demonstrate this speedup, or is it computationally infeasible even for small problems to evaluate the non-pre-computed version?
- In Tables 1 and 4, INSTINCT seems to perform quite well and the authors report average rank, which is fine. For the sake of differntiating between the performance of the two baselines (APE and InstructZero), it would also be interesting to see this summary via performance profiles curves [1, 2], as they are computed over a decently large set of tasks. The caveat is that some of the scores appear to be perfect, which is not directly supported by performance profiles, but there are ways of dealing with this such as setting a performance ceiling.
- The text in figure 4 is too small.

[1] https://arxiv.org/abs/cs/0102001
[2] https://www.argmin.net/2018/03/26/performance-profiles/

---

> ### Author Response · Authors · 2023-11-18
> **Response to Reviewer zDza - Part 1/3**
>
> We would like to thank the reviewer for taking the time to review our paper, and for appreciating that our work has solid contributions, solves a clear problem, and provides thorough evaluations.
>
> ---
>
> > "It is interesting that the method can be sped up via pre-computation. Can the authors demonstrate this speedup"
>
> As suggested, we have conducted additional experiments to compare the running time of our INSTINCT algorithm with and without the pre-computation of the representations. The results (Response Table 1 below) show that **the pre-computation dramatically reduces the running time of our INSTINCT algorithm**.
>
> *Response Table 1: The running time (± standard error) of our INSTINCT algorithm with and without pre-computation. The results are averaged on 20 instruction induction tasks in Table 1 of our paper.*
>
> |    Algorithm         |  Running time (165 iterations) |  Running time (500 iterations) |  Running time (1000 iterations) |  Running time (2000 iterations) |
> |:-------------|------:|------:|------:|------:|
> | Without pre-computation |  77.71 mins (±6.03 mins) | 270.88 mins (±5.11 mins)| 541.76 mins (±10.21 mins) | 1083.52 mins (±20.42 mins) |
> | INSTINCT (with pre-computation)    |  29.05 mins (±0.54 mins) | 80.13 mins (±0.17 mins) | 156.37 mins (±0.24 mins) | 308.86 mins (±0.38 mins) |
> | Speedup | 2.68 times | 3.38 times | 3.46 times | 3.50 times |
>
> In Response Table 1 above, "Without pre-computation" is the baseline whose only difference with our INSTINCT algorithm is that this baseline uses forward passes to compute every hidden representation in each iteration, and "INSTINCT (with pre-computation)" is our algorithm. The first column in Response Table 1 (165 iterations, same as the experiments in our paper) shows that pre-computation drastically reduces the running time of our INSTINCT algorithm from 78 mins to 29 mins. In addition, Response Table 1 also shows that in tasks that require more iterations, the pre-computation brings even more significant speed-ups. This is because as the number of iterations increases, the cost of pre-computation is further amortized across more iterations, which makes the computational savings offered by pre-computation more pronounced.
>
> We agree with the reviewer that the added empirical evaluation of the computational costs (Response Table 1) can further strengthen the contribution of our method of pre-computation. Thank you for pointing this out, and we will add the results and discussions here to the revised paper.
>
> >"Can the authors provide an empirical comparison between the computation costs involved with running INSTINCT compared to its baselines? Ideally for the sake of a fair evaluation, this should also include pre-computation."
>
> As suggested, we have performed additional experiments to empirically compare the running time of InstructZero and our INSTINCT. The results (Response Table 2 below) show that **our INSTINCT has comparable computational costs with InstructZero**.
>
> *Response Table 2: The running time (± standard error) of InstructZero and our INSTINCT. The results are averaged over the 20 instruction induction tasks in Table 1 of our paper.*
>
> |    Algorithm         |   Total runing time |   Time for querying ChatGPT |
> |:-------------|------:|-------:|
> | InstructZero |  18.00 mins (±0.41 mins) |   11.97 mins (±0.35 mins) |
> | INSTINCT (including pre-computation)          |  29.05 mins (±0.54 mins) |   11.99 mins (±0.32 mins) |
>
> Response Table 2 shows that the running time of the two algorithms are on the same scale. Our INSTINCT is slightly slower than InstructZero due to the use of the neural network and the pre-computation of the transformer representations.
> However, with this slight increase in the computational cost, our INSTINCT has achieved significantly better performances than InstructZero as demonstrated in Tables 1, 2, 3, and 4 of our paper. Moreover, as we have shown in our response above (Response Table 1), the method of pre-computation is crucial for our INSTINCT to achieve a comparable computational cost with InstructZero, and the benefit of the pre-computation becomes more significant as a larger query budget is adopted. This further demonstrates the importance of the contribution of our method of pre-computation, and we will also add the results and discussions here to the paper after revision.
>
> ---
> ↓↓↓ Continue below ↓↓↓

---

> ### Author Response · Authors · 2023-11-18
> **Response to Reviewer zDza - Part 2/3**
>
> > "Only compared to two baselines: APE and InstructZero. I do see that this is addressed in Section 6 -- but can any of the black-box methods be adapted to your setting in some simple way?"
>
> To further demonstrate our superior performance over other black-box methods, we have added another comparison with a **concurrent work** (released on arXiv only on Sep 15, 2023): EvoPrompt (Guo et al., 2023) which is based on evolutionary algorithms. The results (Response Table 3 below) show that **our INSTINCT significantly outperforms the concurrent work of EvoPrompt, even though EvoPrompt requires more queries to ChatGPT** (more details below).
>
> *Response Table 3: Test accuracy of the best instructions from different algorithms.*
>
> | Algorithm | APE | InstructZero | EvoPrompt | INSTINCT |
> |:-------------|------:|-------:|------:|----------------------:|
> |antonyms|0.6367(0.1416)|0.8267(0.0072)|0.7967(0.0152)|**0.8467(0.0027)**|
> |auto_categorization|0.2500(0.0094)|0.2567(0.0119)|**0.2600(0.0309)**|0.2500(0.0330)|
> |auto_debugging|0.2917(0.0340)|**0.3750(0.0000)**|**0.3750(0.0000)**|0.2917(0.0340)|
> |cause_and_effect|0.5733(0.0891)|0.8133(0.0109)|**0.8267(0.0475)**|0.5867(0.0871)|
> |common_concept|0.0691(0.0207)|0.0864(0.0398)|0.1211(0.0000)|**0.2129(0.0019)**|
> |diff|0.6733(0.2667)|0.6933(0.2224)|**1.0000(0.0000)**|**1.0000(0.0000)**|
> |informal_to_formal|0.5736(0.0026)|0.5310(0.0024)|**0.6182(0.0047)**|0.5534(0.0000)|
> |letters_list|**1.0000(0.0000)**|0.5900(0.1674)|**1.0000(0.0000)**|**1.0000(0.0000)**|
> |negation|0.7533(0.0109)|0.7767(0.0136)|0.7867(0.0027)|**0.8167(0.0027)**|
> |object_counting|**0.3633(0.0191)**|0.3600(0.0929)|0.1167(0.0626)|0.3400(0.0698)|
> |odd_one_out|0.6333(0.0144)|0.6133(0.0871)|0.6533(0.0109)|**0.7000(0.0163)**|
> |orthography_starts_with|0.4567(0.1477)|0.5067(0.0871)|0.6000(0.0205)|**0.6667(0.0272)**|
> |rhymes|0.1567(0.0640)|**1.0000(0.0000)**|0.6133(0.0178)|**1.0000(0.0000)**|
> |second_word_letter|**0.7467(0.2028)**|0.4333(0.1872)|0.4133(0.2397)|0.1000(0.0411)|
> |sentence_similarity|0.0000(0.0000)|0.0000(0.0000)|**0.2833(0.0357)**|0.1400(0.0047)|
> |sum|0.6733(0.2667)|**1.0000(0.0000)**|**1.0000(0.0000)**|**1.0000(0.0000)**|
> |synonyms|**0.3600(0.0759)**|0.2767(0.0925)|0.1367(0.0054)|0.3067(0.0491)|
> |taxonomy_animal|0.3467(0.2341)|0.7167(0.0838)|0.7167(0.1308)|**0.8567(0.0599)**|
> |word_sorting|0.3300(0.0374)|0.3100(0.1143)|**0.5167(0.0435)**|0.5133(0.0027)|
> |word_unscrambling|0.4400(0.1389)|0.5500(0.0170)|0.6033(0.0072)|**0.6333(0.0072)**|
> | -- | -- | -- | -- | -- |
> | Average ranking | 3.05 | 2.6 | 1.95 | 1.8|
>
> Response Table 3 shows that our INSTINCT consistently outperforms all baselines, including the concurrent work of EvoPrompt.
> In addition, note that EvoPrompt uses more queries to ChatGPT than our INSTINCT. This is because, in every iteration, Evoprompt needs to query ChatGPT to generate a new instruction and query ChatGPT again to obtain its score, whereas our INSTINCT only needs the latter query to ChatGPT (to obtain the score). Despite getting less feedback from the black-box LLM (i.e., ChatGPT), our INSTINCT still performs better than EvoPrompt. This is because our INSTINCT is based on neural bandits which is a **global optimization algorithm** that is able to utilize the historical data to efficiently balance exploration vs. exploitation. In contrast, EvoPrompt uses the evolutionary algorithm which is a **local optimization algorithm** that cannot utilize historical data to efficiently explore the global search space of instructions.
> Moreover, some other concurrent works on black-box methods for instruction optimization are based on similar core ideas as EvoPrompt, i.e., they also use a powerful LLM (e.g., ChatGPT) to propose new candidate instructions via rephrasing and use evolutionary algorithms for instruction optimization.
> Therefore, our performance advantage over the representative method of EvoPrompt is also likely to hold for the other black-box methods.
>
> On the other hand, our INSTINCT is in fact not mutually exclusive with these methods that use LLM to generate instructions, in the sense that **these methods can be incorporated into our INSTINCT algorithm to further improve our performance**. Specifically, as a proof-of-concept experiment, in the last paragraph of Sec. 5 in our paper, we had proposed a method to further improve our INSTINCT via ChatGPT rephrasing (similar to the rephrasing approaches used in APE and EvoPrompt), and Figure 5 shows that this rephrasing method has further improved the performance of our INSTINCT.
>
> ---
> ↓↓↓ Continue below ↓↓↓

---

> ### Author Response · Authors · 2023-11-18
> **Response to Reviewer zDza - Part 3/3**
>
> >  "It would also be interesting to see this summary via performance profiles curves [1, 2], as they are computed over a decently large set of tasks"
>
> Thank you very much for the suggestion and for providing the references.
> We have followed your suggestion to plot the performance profile curves of different algorithms (including the newly added baseline of EvoPrompt discussed in our response above): See https://postimg.cc/McjLP25P (or the same figure at https://pasteboard.co/fvgFXjDuvZ1t.png as backup). The figure shows that **the performance profile curves indeed make it much easier to distinguish the performances of the baselines of APE and InstructZero**, and more importantly, **the performance profile curves also make it easier to visualize the performance superiority of our INSTINCT algorithm**. For ease of reference, we summarize the performance profile curves in Response Table 4 below for your convenience:
>
> *Response Table 4: $\rho(\tau)$ under different $\tau$ for all methods. Higher is better.*
>
> |     $\tau$          |   0.0 |   0.05 |   0.1 |   0.15 |   0.2 |   0.25 |   0.3 |   0.35 |
> |:-------------|------:|-------:|------:|-------:|------:|-------:|------:|----------------------:|
> | APE          |  0.2  |    0.3 |  0.45 |    0.5 |  0.6  |   0.7  |  0.8  |                  0.9  |
> | InstructZero |  0.15 |    0.4 |  0.6  |    0.7 |  0.75 |   0.8  |  0.85 |                  0.95 |
> | EvoPrompt    |  0.45 |    0.6 |  0.75 |    0.8 |  0.8  |   0.9  |  0.9  |                  0.95 |
> | INSTINCT (ours)     |  **0.55** |    **0.7** |  **0.85** |    **0.9** |  **0.9**  |   **0.95** |  **0.95** |                  **0.95** |
>
> We explain here how we have plotted the performance profile curves following the references you have provided. To draw the performance profile curve for a method, for each task $i$, we check whether the performance of this method in task $i$ is within $\tau$ distance to the best performance (among different methods) in task $i$, and hence define an indicator function $\mathbb{I}()$. Next, we average this indicator function across all $n_p$ tasks, which yields a value $\rho(\tau)$ (see equation below). Finally, the performance profile curve for this method is obtained by varying the value of $\tau$ and calculating the corresponding $\rho(\tau)$.
>
> $$\rho(\tau) = \frac{\sum_{i=1}^{n_p}\mathbb{I}\bigl((\text{Best performance in task } i - \text{Performance of this method in task } i) \le \tau \bigl)}{n_p}.$$
>
>
> We have added the performance profile curves as Figure 16 in our revised paper, and will add the figure to the main paper in our future revision. Thank you again for the insightful and helpful suggestion.
>
>
> ---
>
> Thank you again for your encouraging and insightful feedback. We hope our clarifications and additional results could help improve your evaluation of our work.

---

> ### Author Response · Authors · 2023-11-21
> **We would like to know if you have any further questions that require additional clarifications.**
>
> Dear Reviewer zDza,
>
> Thank you again for your time in reviewing our paper and for asking many valuable questions.
>
> Please let us know if our replies have addressed your concerns. We would be happy to address any further questions you might have within the discussion period.
>
>
>
> Best wishes,
>
> Authors

---

> ### Comment · Reviewer_zDza · 2023-11-22
> **Thank you for response!**
>
> Thanks for the thorough response and the additional analyses! These additional results are convincing to me and address all of the comments that I had, so I am happy to raise my score to an 8.

---

> > ### Author Response · Authors · 2023-11-22
> > **Thanks to Reviewer zDza**
> >
> > Dear Reviewer zDza,
> >
> > Thank you so much for your positive feedback! We are happy that we have addressed your concerns and we are deeply encouraged by your recognition of our work.

---

### Official Review · Reviewer_TRUy · 2023-11-05

**Soundness:** 2 fair
**Presentation:** 3 good
**Contribution:** 2 fair
**Rating:** 5
**Confidence:** 5

**Summary:**

This paper introduces  INSTINCT algorithm to optimize the instructions for black-box LLMs. INSTINCT replaces the GP surrogate in BO by an NN surrogate, and couples the NN surrogate with the hidden representation learned by a pre-trained transformer

**Strengths:**

The paper is well-written and easy to follow, and it proposes a novel method to solve the instruction optimization problem.

**Weaknesses:**

1. Clarification is required on one aspect: it appears that the MLP on top the pre-trained model remains unchanged, with its parameters being pre-determined using 1000 pairs of vectors and score. An interesting baseline for the author to consider might be utilizing these Sobol sequences to create instructions and selecting the best one. I'm curious to see if the author's approach would yield any significant improvements through exploration.

2. The author's exploration into whether using ChatGPT for paraphrasing enhances instruction quality raises an important question. If paraphrasing indeed improves instructions, is the optimization process still necessary? Perhaps a more straightforward approach would be to initially generate a rudimentary base instruction using a white-box model, potentially of lower quality, and then refine it through repeated paraphrasing.

**Questions:**

See the weakness part.

---

> ### Author Response · Authors · 2023-11-18
> **Response to Reviewer TRUy - Part 1/3**
>
> We would like to thank the reviewer for taking the time to review our paper, for appreciating our contributions and acknowledging the novelty of our approach, and particularly for highlighting the strengths of our paper in terms of clarity of writing.
>
> ---
>
> > "It appears that the MLP on top of the pre-trained model remains unchanged, with its parameters being pre-determined using 1000 pairs of vectors and score."
>
> We would like to clarify that in our algorithm, **the MLP on top of the pre-trained model is changed (re-trained) after every iteration** and **we do not pre-determine the parameters of the MLP using 1000 vector-score pairs**. We explain in more detail below.
>
> As illustrated in Figure 2 of our paper, in each iteration, we select the soft prompt $z_t$ by maximizing the acquisition function (step 2), use $z_t$ to generate the instruction $\rho_t$ (step 3), and evaluate the score of this instruction $\rho_t$ to produce a score $h_t$ (step 4 and 5). After that, we add the hidden representation $z'_t$ of the soft prompt $z_t$ and the score $h_t$ into the training dataset of the MLP to form a new dataset, and **use this new dataset to retrain the MLP**. Therefore, our MLP is changed (re-trained) after every iteration.
>
> As for pre-training the MLP using the initial dataset, we, in fact, use 40 (instead of 1000) pairs of vectors and scores as the initial dataset (selected using a Sobol sequence) to pre-train the MLP, and subsequently use our algorithm to select further queries (as described above). More specifically, we construct the initial dataset by selecting 40 initial queries (soft prompts) using a Sobol sequence and evaluating these 40 soft prompts using ChatGPT to obtain their scores, after which we use these 40 vector-score pairs to pre-train the MLP. Given that our overall query budget is only 165, we subsequently use our algorithm to sequentially select the remaining 165-40=125 queries, during which we re-train the MLP after every iteration (as described above).
>
> ---
> ↓↓↓ Continue below ↓↓↓

---

> ### Author Response · Authors · 2023-11-18
> **Response to Reviewer TRUy - Part 2/3**
>
> > "An interesting baseline for the author to consider might be utilizing these Sobol sequences to create instructions and selecting the best one."
>
> As clarified in our response above, we select 40 initial queries using the Sobol sequence to generate 40 instructions, and then evaluate these 40 instructions using ChatGPT to obtain their scores. To see how much our INSTINCT algorithm outperforms the initial Sobol sequence, we show in Response Table 1 below (in terms of test accuracy) that **our INSTINCT algorithm significantly improves over the best instruction among these 40 initial instructions selected using the initial Sobel sequence**. Note that validation accuracy (not test accuracy reported in the table) is the metric used during instruction optimization, which explains why in a few tasks, the best instruction among the 40 initial instructions has a higher test accuracy than our INSTINCT algorithm.
>
> *Response Table 1: Test accuracy of the best instruction generated by 40 random points from the Sobol sequence and the instruction generated by INSTINCT.*
>
> |    Tasks          |   40 random Sobol points |   INSTINCT (ours) |
> |:-------------|------:|----------------------:|
> |antonyms| **0.8500** | 0.8467|
> |auto_categorization|0.0100 | **0.2500**|
> |auto_debugging |0.2500| **0.2917**|
> |cause_and_effect |0.5200 |**0.5867**|
> |common_concept |0.0045| **0.2129**|
> |diff| **1.0000**| **1.0000**|
> |informal_to_formal| 0.5394| **0.5534**|
> |letters_list| **1.0000**| **1.0000**|
> |negation| 0.7600| **0.8167**|
> |object_counting| 0.2200| **0.3400**|
> |odd_one_out| 0.6400| **0.7000**|
> |orthography_starts_with| 0.4400| **0.6667**|
> |rhymes| 0.4900| **1.0000**|
> |second_word_letter| **0.1300**| 0.1000|
> |sentence_similarity| 0.0000| **0.1400**|
> |sum| **1.0000**| **1.0000**|
> |synonyms| **0.4200**| 0.3067|
> |taxonomy_animal| **0.9300**| 0.8567|
> |word_sorting| 0.0400| **0.5133**|
> |word_unscrambling| 0.5800| **0.6333**|
> | --- | --- | ---|
> |**# best-performing tasks**| 7| 16|
>
> It is natural for one to ask: How does our INSTINCT compare to a pure random exploration baseline which spends all its 165 query budget using a Sobol sequence? We have conducted further experiments to demonstrate below in Response Table 2 that **our INSTINCT consistently outperforms this pure exploration baseline**. This is because **our INSTINCT effectively balances exploration vs. exploitation** during the optimization process.
>
> *Response Table 2: Test accuracy of the best instruction generated by 165 iterations of random selection from the Sobol sequence (same as the total evaluation budget) and the instruction generated by INSTINCT.*
>
> | Task           | 165 pure random selection  | INSTINCT (ours) |
> |:---------------------------|--------------------------:|--------------------------:|
> | antonyms                  | 0.8067           | **0.8467** |
> | auto\_categorization      | 0.2033           | **0.2500** |
> | auto\_debugging           | **0.2917**       | **0.2917** |
> | cause\_and\_effect        | 0.5467           | **0.5867** |
> | common\_concept           | 0.0196           | **0.2129** |
> | diff                      | 0.2533           | **1.0000** |
> | informal\_to\_formal      | 0.4200           | **0.5534** |
> | letters\_list             | **1.0000**       | **1.0000** |
> | negation                  | 0.6733           | **0.8167** |
> | object\_counting          | **0.3533**       | 0.3400     |
> | odd\_one\_out             | 0.6733           | **0.7000** |
> | orthography\_starts\_with | 0.5233           | **0.6667** |
> | rhymes                    | 0.6000          | **1.0000** |
> | second\_word\_letter      | **0.1167**       | 0.1000     |
> | sentence\_similarity      | 0.0133           | **0.1400** |
> | sum                       | 0.9433           | **1.0000** |
> | synonyms                  | 0.2300           | **0.3067** |
> | taxonomy\_animal          | **0.9367**       | 0.8567     |
> | word\_sorting             | 0.2400           | **0.5133** |
> | word\_unscrambling        | 0.5867           | **0.6333** |
> | --- | --- | ---|
> |**# best-performing tasks**| 5| 17|
>
>
> ---
> ↓↓↓ Continue below ↓↓↓

---

> ### Author Response · Authors · 2023-11-18
> **Response to Reviewer TRUy - Part 3/3**
>
> > "If paraphrasing indeed improves instructions, is the optimization process still necessary?" and "a more straightforward approach would be to initially generate a rudimentary base instruction ... and then refine it through repeated paraphrasing."
>
> We would like to thank the reviewer for pointing out the insights on utilizing paraphrasing and proposing a straightforward approach with repeated rephrasing. Interestingly, **your proposed approach coincides with the main idea of the baseline of APE (explained in Sec. 6)**. We have extensively compared our INSTINCT with APE and **shown that our INSTINCT consistently outperforms APE**. Specifically, the baseline APE performs instruction optimization using the core idea of iterative paraphrasing. APE starts with LLM-generated initial candidate instructions and generates multiple candidates using ChatGPT paraphrasing. After that, APE evaluates the performances of all candidates and keeps the best one while discarding the others. This process is repeated for multiple iterations to find the best instruction. **Figure 1, Table 1 and Table 4 in our paper show that our INSTINCT performs significantly better than APE which is based on paraphrasing**. We will revise our paper to clearly describe APE in the first paragraph of the Experiments section (Sec. 4).
>
> The reason why our INSTINCT is better than the paraphrasing-based approach is that **our INSTINCT is a global optimization algorithm while the paraphrasing-based approach is a local optimization algorithm**. Specifically, our INSTINCT uses neural bandits which utilizes the historical queries to proactively control the exploration-exploitation trade-off and hence can find the global optimum efficiently (i.e., using a small number of queries). In contrast, the paraphrasing-based approach is a local search algorithm that is not able to effectively balance the exploration-exploitation trade-off, as a result, it is unable to efficiently traverse the global search space of instructions and hence cannot find the global optimum efficiently. Therefore, the optimization process is still necessary for achieving the best performance.
>
> ---
>
> Thank you again for your time and your careful feedback. We hope our clarifications and additional results could improve your opinion of our work.

---

> ### Author Response · Authors · 2023-11-21
> **We would like to know if you have any further questions that require additional clarifications.**
>
> Dear Reviewer TRUy,
>
> Thank you again for your time in reviewing our paper and for asking many valuable questions.
>
> Please let us know if our replies have addressed your concerns. We would be happy to address any further questions you might have within the discussion period.
>
>
>
> Best wishes,
>
> Authors

---

> ### Author Response · Authors · 2023-11-23
> **Last day rebuttal reminder: Thank you for reviewing our paper, we would like to know whether our responses have addressed your concerns**
>
> Dear Reviewer TRUy,
>
> Today is the last day of rebuttal, and we are eager to know whether our responses have addressed your concerns. To summarize:
>
> 1. We conduct additional experiments to show our INSTINCT improves significantly over random Sobol sequences.
> 2. We compare with the paraphrasing-based algorithm (our INSTINCT is much better) and explain why our optimization process is still necessary.
>
> We are happy to address any further questions you might have before the rebuttal ends.
>
> Best wishes,
>
> Authors

---

### Author Response · Authors · 2023-11-22
**Summarization of our rebuttal**

Dear Reviewers,

Thanks for your detailed review. However, we have not received any responses from you during the rebuttal. We wish to know whether our response has answered the questions raised by you and are happy to provide further clarifications during the discussion period. In the rebuttal, we have provided additional and extensive experiments to answer the questions from you. Specifically, we have made the following major clarifications:
1. The comparison with a random sampling baseline in the response to Reviewer TRUy.
2. The analysis and experiments on the running time of our algorithm to show the speedup of the pre-computation technique in our algorithm and comparison of running time with other baselines in the response to Reviewer zDza.
3. The comparison of performance with a concurrent work that uses evolutionary algorithms to do instruction optimization in the response to Reviewer zDza and Reviewer fbJ6.
4. The comparison of the performance of our algorithm under different combinations of black-box and white-box LLMs (which are from different families) in the response to Reviewer 16Xs and Reviewer fbJ6.
5. The comparison of the performance of our algorithm and other baselines under different dimensionalities of soft prompts in response to Reviewer fbJ6.
6. The reason why we use neural bandits instead of other BO algorithms in the response to Reviewer fbJ6.
7. The reason why our INSTINCT algorithm is better than other local search algorithms (e.g., the evolutionary algorithm, paraphrasing-based algorithms) in response to Reviewer TRUy and Reviewer fbJ6.

We wish to thank the reviewers again for the detailed feedback. We hope our response will improve your opinions of our work.

Best wishes,

Authors

---

### Meta-Review · Area_Chair_x7zp · 2023-12-21

**Metareview:**

After careful consideration of the reviewers' feedback and the authors' responses during the rebuttal phase, the decision is to reject this paper. Although the authors have made substantial efforts to address the concerns raised by the reviewers, there are still unresolved issues that impact the overall acceptance of the work.

The primary reason for this decision is the lack of novelty. The proposed INSTINCT algorithm, while an interesting approach to instruction optimization for LLMs, does not sufficiently differentiate itself from existing methodologies. The combination of InstructZero and NeuralUCB, despite the authors' extensive rebuttal, has not convinced the reviewers that the method constitutes a significant leap in the state-of-the-art.

Additionally, the concerns about the evaluation on different combinations of white-box and black-box models, though partially addressed in the rebuttal, still leave questions about the generalizability of the algorithm across different LLM families. While the authors have provided additional experiments, the results are not compelling enough to suggest that INSTINCT is a universally applicable solution.

Another point of contention is the choice of the neural bandit approach over other modern BO strategies. The reviewers have indicated that a comparison with a wider range of modern BO strategies or evolutionary algorithms might have provided a stronger case for the superiority of INSTINCT. The authors' rebuttal provides some additional comparisons but falls short of a comprehensive benchmarking that would be required to establish the distinct advantages of their approach.

The authors' rebuttal is appreciated for its depth and the additional experimental results provided. However, the concerns regarding novelty, evaluation breadth, and methodological justification remain significant enough to suggest that the paper is not ready for acceptance at this time.

**Justification For Why Not Higher Score:**

The decision to reject rather than offer a major revision or accept is based on the balance of the strengths and weaknesses of the paper, as well as the authors' ability to address the core concerns during the rebuttal phase. While the authors have made commendable efforts to provide additional experiments and clarifications, these do not fully overcome the fundamental issues highlighted by the reviewers.

**Justification For Why Not Lower Score:**

N/A

---

### Decision · Program_Chairs · 2024-01-16

Reject